# Diffusion Pretraining for Gait Recognition in the Wild

## Abstract

Recently, diffusion models have garnered much attention for their remarkable generative capabilities. Yet, their application for representation learning remains largely unexplored. In this paper, we explore the possibility of using the diffusion process to pretrain the backbone of a deep learning model for a specific application—gait recognition in the wild. To do so, we condition a latent diffusion model on the output of a gait recognition model backbone. Our pretraining experiments on the Gait3D and GREW datasets reveal an interesting phenomenon: diffusion pretraining causes the gait recognition backbone to separate gait sequences belonging to different subjects further apart than those belonging to the same subjects, which translates to a steady improvement in gait recognition performance. Subsequently, our transfer learning experiments on Gait3D and GREW show that the pretrained backbone can serve as an effective initialization for the downstream gait recognition task, allowing the gait recognition model to achieve better performance within much fewer supervised training iterations. We validated the applicability of our approach across multiple existing gait recognition methods and conducted extensive ablation studies to investigate the impact of different pretraining hyperparameters on the final gait recognition performance.

## 1 Introduction

Gait, the unique manner in which a person walks, offers an alternative way to identify individuals, alongside the more common biometric modalities like fingerprints and irises. With the emergence of deep learning, advancements in computer vision architectures, and the collection of well-labelled gait datasets, deep gait recognition has become an increasingly popular area of research over the last few years (Sepas-Moghaddam & Etemad, 2022). Consequently, owing to the efforts of many previous works, it has become possible to achieve impressive accuracy performance on existing controlled datasets such as CASIA-B (Yu et al., 2006) and OU-MVLP (Takemura et al., 2018). However, when these techniques are applied to recently released in-the-wild gait datasets such as Gait3D (Zheng et al., 2022) and GREW (Zhu et al., 2021), their performance pales in comparison, highlighting their limited applicability to unconstrained settings. As a result, recent studies (Zheng et al., 2022; Zhu et al., 2021; Lin et al., 2022; Fan et al., 2023c; Cosma et al., 2023; Fan et al., 2023a; Habib et al., 2024) have begun shifting their focus towards addressing the more challenging problem of gait recognition in the wild.

Recently, diffusion models (Sohl-Dickstein et al., 2015; Ho et al., 2020; Song et al., 2020; Nichol & Dhariwal, 2021; Rombach et al., 2022; Song et al., 2023) have been in the spotlight for their impressive performance on generative tasks, outshining variational autoencoders, flow-based generative models, and generative adversarial networks which previously dominated the generative realm. Their stability during training and ability to generate high-quality realistic samples, such as images (Rombach et al., 2022; Nichol et al., 2021; Ramesh et al., 2022; Saharia et al., 2022; Dhariwal & Nichol, 2021), videos (Ho et al., 2022b;a; Singer et al., 2022; Bar-Tal et al., 2024), and audio (Schneider, 2023; Huang et al., 2023; Mittal et al., 2021), have led to their wide adoption in academia and industry. Amidst the hype to exploit diffusion models for generative tasks, other promising directions of diffusion models have been less explored. One such direction involves leveraging diffusion models for representation learning, where a model learns to extract relevant features during the diffusion process, which can be beneficial for tasks beyond generation. Moreover, with a simple reconstruction or noise prediction objective, diffusion training can potentially serve as an

effective self-supervised pretraining approach, which does not necessitate the need for a labelled dataset. While some studies have started looking into using the learnt representations for common tasks like image classification (Clark & Jaini, 2024; Hudson et al., 2023; Li et al., 2023; Xiang et al., 2023; Abstreiter et al., 2021; Yang & Wang, 2023) and segmentation (Yang & Wang, 2023; Zhao et al., 2023), not much attention has been paid to using them in applications such as gait recognition. Furthermore, many previous works related to diffusion-based representation learning have not investigated the effects of finetuning the learnt representations on downstream tasks, overlooking the pretraining potential of diffusion training.

Considering the limited research on diffusion-based representation learning in the gait recognition field and the more challenging in-the-wild scenarios, we aim to address the following question: can we leverage diffusion training to enhance existing methods for gait recognition in the wild?

To explore this, we propose a diffusion-based approach to pretrain the backbone of a gait recognition model by using its output as a condition for a latent diffusion model. Thereafter, we initialize the gait recognition model with the pretrained backbone and perform transfer learning on the downstream gait recognition task. We conducted extensive applicability studies with multiple existing gait recognition models, including GaitGL (Lin et al., 2022), GaitPart (Fan et al., 2020), GaitSet (Chao et al., 2019), SMPLGait w/o 3D (Zheng et al., 2022), and GaitBase (Fan et al., 2023c), by evaluating on two in-the-wild gait datasets, Gait3D (Zheng et al., 2022) and GREW (Zhu et al., 2021). Additionally, we performed thorough ablation studies to investigate the effects that different pretraining hyperparameters have on the downstream gait recognition task.

Our finding reveals that during diffusion pretraining, the gait recognition model backbone, regardless of its architecture, learns to separate gait sequences belonging to different subjects further apart than those belonging to the same subject, despite the lack of an explicit signal to do so. This results in a steady improvement in the gait recognition performance and demonstrates the potential of diffusion pretraining for gait recognition. Subsequently, when the gait recognition model is initialized with the pretrained backbone and further finetuned on the downstream gait recognition task, it surpasses the performance of its trained-from-scratch counterpart by as much as 7.9% on Gait3D and 4.2% on GREW. This remains the case even when the number of supervised training iterations is significantly reduced by as much as 89% on Gait3D and 70% on GREW.

To the best of our knowledge, we are the first to apply diffusion training for representation learning in the field of gait recognition and demonstrate its effectiveness as a pretraining approach for the gait recognition task.

## 2 RELATED WORK

### 2.1 GAIT RECOGNITION

With the advent of deep learning, gait recognition typically involves extracting gait features from gait sequences and projecting these features into more discriminative embeddings that can be compared using a distance metric such as Euclidean or cosine distances. These gait sequences typically come in the form of either silhouettes or skeletons, with silhouette-based recognition being more popular (Sepas-Moghaddam & Etemad, 2022), though skeleton-based gait recognition remains an active area of research (Teepe et al., 2021; 2022; Zhang et al., 2023).

Much state-of-the-art research regarding deep gait recognition focuses on the architectural design of backbone networks to maximize the extraction of meaningful gait information from gait sequences (Lin et al., 2022; Fan et al., 2023c; Cosma et al., 2023; Fan et al., 2023a; Chao et al., 2019; Fan et al., 2020; Song et al., 2019). For instance, GaitPart (Fan et al., 2020) proposes a part-based approach that divides each gait silhouette into several parts and extracts features for each part separately to obtain more fine-grained features. It also focuses on short-range neighbouring frames rather than considering all frames within a gait sequence. GaitSet (Chao et al., 2019), on the other hand, treats gait as an unordered set of gait silhouettes and uses only permutation-invariant operations within its architecture.

Inspired by the fields of person reidentification, current state-of-the-art research (Chao et al., 2019; Fan et al., 2020; Lin et al., 2022; Zheng et al., 2022; Fan et al., 2023c) also adapted techniques such as horizontal pyramid matching (Fu et al., 2019) and batch normalization neck (Luo et al., 2019) to

further enhance gait recognition accuracy. Several works have also explored self-supervised learning approaches (Liu et al., 2021; Rao et al., 2021; Fan et al., 2023b; Cosma et al., 2023). They are often based on contrastive learning, where augmented versions of the same gait sequences are treated as positive pairs and different gait sequences are treated as negative pairs.

While many of these methods have performed well on existing controlled datasets such as CASIA-B (Yu et al., 2006) and OU-MVLP (Takemura et al., 2018), their performance drastically drops when applied to the recently released in-the-wild datasets (Zheng et al., 2022; Zhu et al., 2021), where factors such as temporary occlusions, varying camera viewpoints, and illumination inadvertently come into play. Finding a way that can universally enhance the performance of these methods in such unconstrained settings is highly desirable.

As the introduction of in-the-wild gait datasets is fairly recent, many earlier works did not have the opportunity to evaluate their methods on these datasets. However, with the efforts of OpenGait (Fan et al., 2023c), several previous works have been reproduced, trained, and evaluated on the in-the-wild datasets, serving as baselines for future research in gait recognition in the wild.

## 2.2 DIFFUSION

First proposed by Sohl-Dickstein et al. (2015), diffusion models, particularly Denoising Diffusion Probabilistic Models (Ho et al., 2020), are now an active area of research. A typical diffusion process consists of two phases—the forward phase and the reverse phase. During the forward phase, a data sample $x$ is gradually transformed into an approximate pure noise by iteratively adding noise $\epsilon \sim \mathcal{N}(\mathbf{0}, \mathbf{I})$ to a data sample across a series of timesteps based on a defined noise schedule $\bar{\alpha}_t$. At any timestep $t$ of the forward diffusion process, the noised data sample $x_t$ can be expressed as:

$$x_t = \sqrt{\bar{\alpha}_t} \cdot x + \sqrt{1 - \bar{\alpha}_t} \cdot \epsilon \tag{1}$$

On the other hand, during the reverse phase, noise is iteratively removed from a noisy sample until a clean sample is obtained. By training a model $\epsilon_\theta$ to predict the added noise $\epsilon$ in the noisy data sample $x_t$ at any given timestep $t$ of the forward diffusion process, a mapping from random noise to the data manifold can be learnt, which confers diffusion models their generative ability. The overall training objective of diffusion models can be summarised as the minimization of the following noise prediction loss $L$ (Ho et al., 2020):

$$L = ||\epsilon - \epsilon_\theta(x_t, t)||_2^2 \tag{2}$$

In order to generate more precise data samples, diffusion models are fed with an additional prompt $c$ and instead learn a conditional data distribution. In this case, the above loss becomes:

$$L = ||\epsilon - \epsilon_\theta(x_t, t, c)||_2^2 \tag{3}$$

Various methods have been proposed to improve the quality of samples produced by conditional diffusion models (Dhariwal & Nichol, 2021; Ho & Salimans, 2022; Wallace et al., 2023). In particular, classifier-free guidance (Ho & Salimans, 2022), which randomly drops out the added condition, has been one of the most simple and effective.

To reduce the computation cost and memory consumption brought about by training with and generating high-resolution data, Rombach et al. (2022) proposed latent diffusion, whereby data are first encoded into a lower dimensional latent space before the diffusion process is applied. To further optimize the training efficiency of diffusion models, various noise schedulers (Ho et al., 2020; Nichol & Dhariwal, 2021; Rombach et al., 2022) and timestep weighting approaches (Hang et al., 2023; Salimans & Ho, 2022) have also been proposed. These methods serve to prioritize a certain noise range during the diffusion process, which improves convergence. Initially started on image generation (Rombach et al., 2022; Song et al., 2023; Nichol et al., 2021; Ramesh et al., 2022; Saharia et al., 2022; Dhariwal & Nichol, 2021), diffusion models have now evolved to generate videos (Ho et al., 2022b;a; Singer et al., 2022; Bar-Tal et al., 2024), audio (Schneider, 2023; Huang et al., 2023; Mittal et al., 2021) and even architecture-specific neural network parameters (Soro et al., 2024).

Despite its impressive generative feats in various domains, the learnt representations produced by the diffusion process have not been looked into as intensively, particularly in specialized fields such as gait recognition. Closely related to our work, Hudson et al. (2023) assessed the usefulness of the

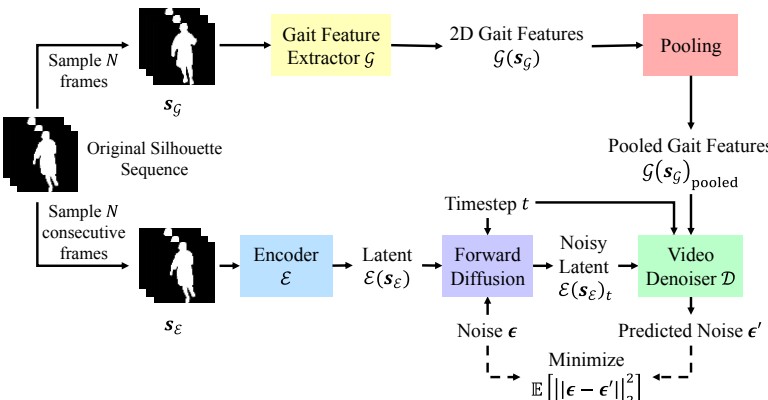

Figure 1: Proposed architecture. Only the gait feature extractor and video denoiser are trained during diffusion pretraining.

representations learnt by training an encoder and a denoiser with only diffusion and evaluating them through linear-probing experiments on image classification tasks. However, the usefulness of the learnt representations in enhancing the performance on downstream tasks has not been looked into.

## 3 PROPOSED METHOD

In this section, we introduce our proposed method, which consists of two stages: (1) pretraining with diffusion and (2) transfer learning on the downstream gait recognition task.

### 3.1 DIFFUSION PRETRAINING

To learn gait representations via diffusion, we employ an end-to-end trainable conditional latent diffusion model. As illustrated in Fig. 1, our architecture consists of three main components—an encoder $\mathcal{E}$, a denoiser $\mathcal{D}$, and a gait feature extractor $\mathcal{G}$. The encoder first encodes an input silhouette sequence into a more compact latent representation, which is then added with random noise and subsequently denoised via the denoiser. At the same time, relevant features from the input silhouette sequence are extracted via the gait feature extractor and these features are pooled and passed on as a condition to aid the denoiser in the denoising process for a more precise reconstruction of the input silhouette sequence. The following sections delve into the individual components that make up the architecture.

#### 3.1.1 ENCODER

Considering its small size and decent silhouette-encoding ability (Appendix A.1), we use the open-sourced Tiny AutoEncoder for Stable Diffusion (TAESD) (Bohan, 2023) as our encoder. As silhouette sequences are made up of single-channel grayscale frames and TAESD only accepts three-channel images, silhouette sequences are first replicated along the channel dimension before encoding. Given a three-channel sequence $\boldsymbol{s}$ of shape $(t, 3, h, w)$, where $t$, $h$, $w$ denote sequence length, height, and width, respectively, the encoder of TAESD, $\mathcal{E}$, performs framewise encoding of the sequence along the spatial dimension while preserving the temporal dimension, producing a $48\times$ compressed latent encoding $\mathcal{E}(\boldsymbol{s})$ of shape $(t, 4, \frac{h}{8}, \frac{w}{8})$.

#### 3.1.2 DENOISER

As for the denoiser, given the lack of open-sourced pretrained gait silhouette sequence diffusion models, we adapt a recent video diffusion model architecture proposed by Ho et al. (2022b) to fit our use case. The denoiser accepts noisy latent representations, selected diffusion timesteps, and any other one-dimensional conditions as inputs, and outputs a prediction of the added noise.

### 3.1.3 GAIT FEATURE EXTRACTOR

The gait feature extractor is our target model to be trained by the diffusion process. To aid the denoiser in predicting the added noise, an additional input condition, corresponding to the gait features of the encoded input sequence into the denoiser, is further provided to the denoiser. Given a silhouette sequence $s$, the gait feature extractor $\mathcal{G}$ transforms it into a two-dimensional gait feature, $\mathcal{G}(s)$, which is representative of the gait contained within the sequence. The gait feature is extracted by the backbone of a typical deep gait recognition model, which can be based on any previous work. To test the applicability of our approach, we adopt a variety of existing backbones in literature to be the gait feature extractor—GaitGL (Lin et al., 2022), GaitPart (Fan et al., 2020), GaitSet (Chao et al., 2019), SMPLGait w/o 3D (Zheng et al., 2022), and GaitBase (Fan et al., 2023c).

### 3.1.4 POOLING METHOD

As the denoiser accepts only one-dimensional tensors as conditions while the extracted gait features are two-dimensional tensors, an operation is needed to convert the gait feature tensors into a single dimension. A pooling operation is applied, rather than simply flattening, to reduce the computation that will be introduced from conditioning. In particular, mean pooling is used, following our ablation study in Sec. 5. The pooled gait feature conditions, $\mathcal{G}(s)_{\text{pooled}}$, are then concatenated with the timestep conditions and are used to scale and bias the activations within the denoiser layers, following the work of Dhariwal & Nichol (2021).

### 3.1.5 INPUT PRETREATMENT

To allow the gait extractor to focus learning on gait information rather than video information, the autoencoder and gait feature extractor are presented with different subsequences sampled from the same silhouette sequence via different sampling algorithms. We denote this pair of input subsequences as $s = (s_{\mathcal{E}}, s_{\mathcal{G}})$, where $s_{\mathcal{E}}$ denotes an input subsequence to the encoder while $s_{\mathcal{G}}$ denotes an input subsequence to the gait feature extractor and $s_{\mathcal{E}} \neq s_{\mathcal{G}}$.

As a video diffusion model is used for the denoiser, its input needs to be temporally consistent and thus, $N$ frames are sampled consecutively from the silhouette sequence to serve as $s_{\mathcal{E}}$. As for $s_{\mathcal{G}}$, we sample $N$ frames using the exact sampling algorithm employed by the authors who proposed the respective gait feature extractor architecture (Lin et al., 2022; Fan et al., 2020; Chao et al., 2019; Zheng et al., 2022; Fan et al., 2023c). We fixed $N = 30$ throughout the study. Even though $s_{\mathcal{E}}$ and $s_{\mathcal{G}}$ are different, $\mathcal{G}(s_{\mathcal{G}})$ should still ideally correspond to the gait features of $s_{\mathcal{E}}$, since $s_{\mathcal{E}}$ and $s_{\mathcal{G}}$ belong to the same subject.

Each frame in $s_{\mathcal{G}}$ is first normalized and resized to $64 \times 64$ before being cropped to $64 \times 44$, which is the common input size for many gait recognition models. As for $s_{\mathcal{E}}$, they are simply normalized and resized to $64 \times 64$.

To increase the generalizability of the model, $s_{\mathcal{E}}$ and $s_{\mathcal{G}}$ are augmented separately with a combination of RandomAffine, RandomPerspective, RandomHorizontalFlip, RandomPartDilate (Fan et al., 2023b) and RandomPartBlur. Appendix A.2 provides a visualization of each augmentation applied.

### 3.1.6 NOISE SCHEDULER AND LOSS WEIGHTING STRATEGY

For generative purposes, noise schedulers, such as the cosine noise scheduler, which prioritize high and low noise levels are commonly adopted. However, in the case of representation learning, Hudson et al. (2023) have found that prioritizing medium noise is more helpful. Following their work, we adopt an inverted cosine noise scheduler. To further focus on medium-level noise, we adopt a medium noise prioritization weighting strategy that downweighs the loss when the noise is too high or too low. Specifically, we modify the existing Min-SNR weighting strategy (Hang et al., 2023), which downweighs losses from low-level noise, to also downweigh losses from high levels of noise.

### 3.1.7 LOSS FUNCTION

To pretrain our gait recognition model via diffusion, we employ the $L2$ noise prediction loss commonly used by diffusion models, which can be summarized as:

$$L_{\text{diffusion}} = ||\epsilon - \mathcal{D}(\mathcal{E}(s_{\mathcal{E}})_t, t, \mathcal{G}(s_{\mathcal{G}})_{\text{pooled}})||_2^2 \tag{4}$$

where $\mathcal{E}(s_{\mathcal{E}})_t$ is the noised latent representation fed to the denoiser $\mathcal{D}$, at timestep $t \in (0, 1000]$ of the forward diffusion process, $\epsilon \sim \mathcal{N}(\mathbf{0}, \mathbf{I})$ is the random noise added, and $\mathcal{G}(s_{\mathcal{G}})_{\text{pooled}}$ is the pooled gait feature condition. During diffusion pretraining, the denoiser and gait feature extractor are trained from scratch while the pretrained TAESD encoder is kept frozen. A higher learning rate is assigned to the gait feature extractor to enable it to better guide the denoiser during training, following Hudson et al. (2023).

## 3.2 Transfer Learning

Once the gait feature extractor is trained by diffusion, we evaluate it on the downstream gait recognition task. We replicate the remaining parts of the gait recognition model accordingly and initialize the weights of the gait backbone with the ones learnt during diffusion pretraining. The untrained parameters are initialized based on the settings provided by OpenGait. We consider two cases of transfer learning, namely, with and without finetuning of the pretrained backbone. During the first case, the pretrained backbone is frozen to evaluate the usefulness of the learnt representations for the downstream gait recognition task. As for the latter case, the pretrained backbone is allowed to be finetuned together with the untrained parameters to evaluate the effectiveness of diffusion as a pretraining method.

During transfer learning, each gait recognition model is trained using the standard triplet loss, $L_{\text{triplet}}$, for identification. The distance measure typically employed is the Euclidean distance. However, we employ the cosine distance instead as we found that the use of cosine distance during training and evaluation produces much better results for existing works (Appendix A.3).

Additionally, some methods, such as GaitBase and SMPLGait w/o 3D, also incorporate an additional smoothed identity loss, $L_{\text{ID}}$, by having another module predict the identity of each gait sequence to enhance the gait recognition performance. In this case, the net loss is the sum of the triplet loss and the reweighted smoothed ID loss:

$$L_{\text{net}} = L_{\text{triplet}} + 0.1 \cdot L_{\text{ID}} \tag{5}$$

## 4 Experiments

### 4.1 Experimental Setup

With the focus on practical gait recognition, two datasets meant for gait recognition in the wild, namely, Gait3D (Zheng et al., 2022) and GREW (Zhu et al., 2021), were chosen. Note that only silhouette sequences from these datasets were used. For pretraining, we used AdamW with warmup and cosine annealing with a learning rate of $1e^{-4}$ for the denoiser and $5e^{-4}$ for the gait feature extractor. The models were trained for 120k iterations with a batch size of 64 for Gait3D and 128 for GREW. More specific hyperparameter settings are detailed in Appendix A.4.

For transfer learning, we trained and evaluated on the same dataset that was used during diffusion pretraining. We used the rank-1 gait recognition accuracy as our main evaluation metric. All transfer learning hyperparameters, unless mentioned in Sec. 4.3, were kept the same as those provided in the OpenGait framework. For GREW, we submitted the results to the official website for evaluation.

To evaluate the effectiveness of our proposed method, we reproduced the results of the corresponding gait recognition models on the Gait3D and GREW datasets trained solely via the supervised objective using the OpenGait framework. Recognizing that data augmentation can enhance the performance of gait recognition models (Fan et al., 2023c), we also included the case when it is applied. The performance of each reproduced baseline is presented in Table 1.

### 4.2 Pretraining Results

To examine if the gait feature extractor has learnt something useful for gait recognition during diffusion pretraining, we measured the gait recognition performance at different checkpoints of the pretraining process (Fig. 2). This was done by feeding the test set's silhouette sequences into the gait feature extractor and directly using either the cosine or Euclidean distance between the output gait features to determine their similarity. For convenience, the accuracy observed during pretraining for GREW was based on the test set defined in OpenGait. Due to space constraints, we only

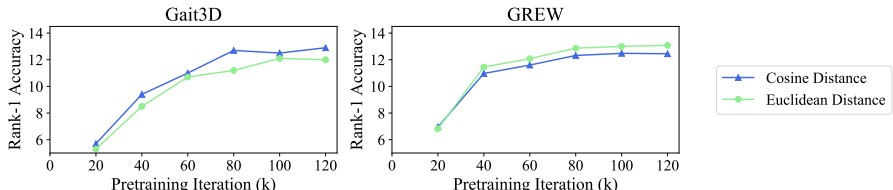

Figure 2: Rank-1 accuracy curves during diffusion pretraining on Gait3D and GREW (GaitSet).

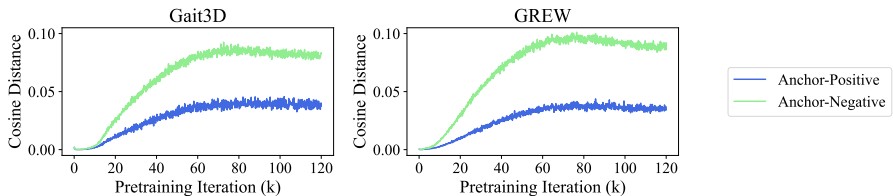

Figure 3: Mean cosine distance of anchor-positive pair and anchor-negative pair during diffusion pretraining on Gait3D and GREW (GaitSet).

show the findings for one of the gait feature extractors, GaitSet, as it was pretrained on Gait3D and GREW. The findings for other methods are available in Appendix A.5.

Interestingly, we observed a steady improvement in gait recognition performance during the diffusion pretraining process, showing that the gait extractor had undoubtedly learnt to extract some features useful for gait recognition. Note that we are simply reconstructing the inputs at this stage.

To investigate further, we recorded the mean cosine distance of the gait features of anchor-positive pairs and the gait features of anchor-negative pairs within a batch during pretraining on Gait3D and GREW (Fig. 3). In our case, an anchor-positive pair refers to an augmented pair $(s_\mathcal{E}, s_\mathcal{G})$ sampled from the same silhouette sequence. In contrast, an anchor-negative pair refers to an augmented pair found within the batch where $s_\mathcal{E}$ and $s_\mathcal{G}$ are sampled from different silhouette sequences.

Looking at Fig. 3, we observe an interesting phenomenon—the difference in the cosine distances between the anchor-positive pairs and anchor-negative pairs increases and stabilises during diffusion pretraining. This is independent of the architecture of the gait feature extractor as well as the dataset, suggesting that our proposed diffusion pretraining approach trains the gait feature extractor to maintain some margin of separation between the anchor-positive pairs and anchor-negative pairs. This is in spite of not having any supervisory signal to encourage separation during training and likely explains why we see an improvement in gait recognition performance even at this stage.

### 4.3 TRANSFER LEARNING RESULTS

#### 4.3.1 FROZEN BACKBONE

With diffusion pretraining, we have pretrained the backbones of various existing gait recognition models. What is left is to project the features into more discriminative embeddings with the remaining parts of these models for the downstream gait recognition task. To evaluate how discriminative the learnt features are for gait recognition, we froze the pretrained backbones and simply trained the remaining untrained layers via the supervised learning objective. The results are shown in Table 1.

Through transfer learning, we achieved higher gait recognition accuracy compared to using the gait features alone. That said, compared to what could be achieved solely by supervised learning on the gait recognition task, the learnt gait features are not as discriminative. This is likely the case since not all the features learnt for input reconstruction are relevant to gait recognition. Nonetheless, some discriminative features are learnt during diffusion training, highlighting its pretraining potential.

#### 4.3.2 FINETUNING OF THE BACKBONE

Next, we allowed the pretrained backbones to be finetuned to investigate if improvements could be made. During finetuning, we found that the ratio of the learning rate of the pretrained layers to

Table 1: Rank-1 accuracy on Gait3D and GREW. GaitPart on GREW is excluded due to instability during training with cosine distance. For train iteration, X + Y denote Xk iterations of diffusion pretraining followed by Yk iterations of transfer learning. Transfer learning iterations for GaitBase on GREW without and with data augmentation are 90k and 120k, respectively.

| Method | Gait3D | | | | GREW | | | |
| | Rank-1 Accuracy (%) | | $r$ | Train Iter. $(\times 10^3)$ | Rank-1 Accuracy (%) | | $r$ | Train Iter. $(\times 10^3)$ |
| | ✗ Data Aug. | ✓ Data Aug. | | | ✗ Data Aug. | ✓ Data Aug. | | |
|---|---|---|---|---|---|---|---|---|
| **Reproduced Baseline** | | | | | | | | |
| GaitGL | 29.2 | 32.4 | - | 180 | 54.0 | 58.4 | - | 250 |
| GaitPart | 31.2 | 38.7 | - | 180 | - | - | - | - |
| GaitSet | 42.2 | 47.8 | - | 180 | 48.1 | 53.1 | - | 250 |
| SMPLGait w/o 3D | 45.5 | 42.9 | - | 180 | 47.6 | 52.1 | - | 250 |
| GaitBase | 56.5 | 65.8 | - | 60 | 58.1 | 61.8 | - | 180 |
| **Diffusion Pretraining + Transfer Learning with Frozen Backbone** | | | | | | | | |
| GaitGL | 17.0 | - | 0.0 | 120 + 60 | 32.2 | - | 0.0 | 120 + 125 |
| GaitPart | 18.8 | - | 0.0 | 120 + 60 | - | - | - | - |
| GaitSet | 23.5 | - | 0.0 | 120 + 60 | 33.5 | - | 0.0 | 120 + 125 |
| SMPLGait w/o 3D | 30.7 | - | 0.0 | 120 + 60 | 36.0 | - | 0.0 | 120 + 125 |
| GaitBase | 35.0 | - | 0.0 | 120 + 60 | 40.5 | - | 0.0 | 120 + 90 |
| **Diffusion Pretraining + Transfer Learning with Finetuning of Backbone** | | | | | | | | |
| GaitGL | 34.4 (↑ 5.2) | 34.4 (↑ 2.0) | 0.1 | 120 + 60 | 56.3 (↑ 2.3) | 58.6 (↑ 0.2) | 1.0 | 120 + 125 |
| GaitPart | 35.7 (↑ 4.5) | 41.7 (↑ 3.0) | 0.5 | 120 + 60 | - | - | - | - |
| GaitSet | 45.0 (↑ 2.8) | 49.9 (↑ 2.1) | 0.5 | 120 + 60 | 52.0 (↑ 3.9) | 55.4 (↑ 2.3) | 1.0 | 120 + 125 |
| SMPLGait w/o 3D | 53.4 (↑ 7.9) | 60.7 (↑ 17.8) | 0.5 | 120 + 60 | 51.8 (↑ 4.2) | 54.1 (↑ 2.0) | 1.0 | 120 + 125 |
| GaitBase | 62.3 (↑ 5.8) | 69.7 (↑ 3.9) | 1.0 | 120 + 60 | 58.5 (↑ 0.4) | 62.0 (↑ 0.2) | 0.5 | 120 + 90/120 |

Table 2: Rank-1 accuracy on GREW with further reduction in supervised training iterations.

| Method | GREW | | |
| | Rank-1 Accuracy (%) | | Train Iter. $(\times 10^3)$ |
| | ✗ Data Aug. | ✓ Data Aug. | |
|---|---|---|---|
| GaitGL | 54.3 | 56.1 | 120 + 75 |
| GaitSet | 51.1 | 53.3 | 120 + 75 |
| SMPLGait w/o 3D | 50.3 | 51.8 | 120 + 75 |

that of the untrained layers, $r$, is an important hyperparameter. We attempted $r \in \{0.1, 0.5, 1.0\}$ for Gait3D and GREW. Moreover, for the GREW dataset, different from the baseline settings with zero weight decay, we added a small weight decay term of $5e^{-5}$ during the finetuning of GaitGL, GaitSet, and SMPLGait w/o 3D as we observed cases of overfitting with the pretrained backbone. Table 1 shows the best finetuning results obtained with the corresponding value of $r$ used. Results for other $r$ values are provided in Appendix A.6.

Looking at Table 1, we observe that all models initialized with the diffusion-pretrained backbone exhibit improved gait recognition performance compared to their trained-from-scratch counterparts, even with a significantly reduced number of supervised training iterations. Excluding the anomalous case for SMPLGait w/o 3D which deteriorated with data augmentation on Gait3D, we observe improvements in rank-1 accuracy by as much as 7.9% and 4.2% on the Gait3D and GREW datasets, respectively. For SMPLGait w/o 3D, which performed poorly with data augmentation on the Gait3D dataset when trained from scratch, initializing its backbone with diffusion-pretrained weights helped to overcome its poor initialization. This underscores the effectiveness of diffusion pretraining as a method to provide a decent initialization point for the downstream gait recognition task.

To further prove our point, we show the rank-1 accuracy curves of the finetuned models with their respective baselines on the Gait3D dataset for the case when no data augmentation is applied during supervised training (Fig. 4). With the backbone pretrained by diffusion, the models outperformed their corresponding supervised baselines within as little as 20k iterations, corresponding to an 89% reduction in supervised training iterations. As for GREW, we show that GaitGL, GaitSet, and SM-PLGait w/o 3D were still able to achieve competitive performance when the supervised training iterations were further reduced to 30% of their baselines' total training iteration (Table 2). Moreover, we show that our final checkpoints demonstrate improved cross-domain performance compared to the respective baselines in Appendix A.7. Clearly, diffusion pretraining provided a good initialization point for the downstream gait recognition task.

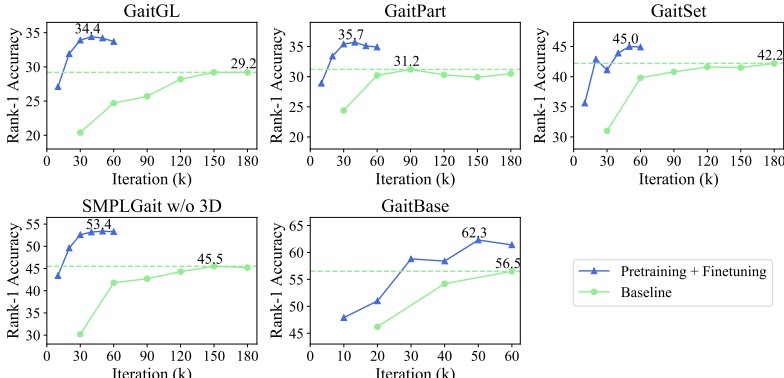

Figure 4: Rank-1 gait recognition accuracy curves when no data augmentation is applied during supervised training (Gait3D).

Table 3: Downstream rank-1 gait recognition accuracy on Gait3D for different diffusion pretraining hyperparameter settings. The first row shows the default setting we used for Gait3D. Highlighted entries denote the different hyperparameter settings compared to our default setting.

| Diffusion Pretraining Hyperparameters | | | | | | | Gait3D |
|---|---|---|---|---|---|---|---|
| Noise Scheduler | Loss Weighting | Pooling | Data Augmentation | Denoiser Size | $l_\mathcal{G}/l_\mathcal{D}$ | $P_{uncond}$ | R-1 (%) |
| Inverted Cosine | MNP | Mean | ✓ | 11.6 M | 5 | 0.15 | 49.1 |
| Inverted Cosine | Uniform | Mean | ✓ | 11.6 M | 5 | 0.15 | 47.6 |
| Cosine | Uniform | Mean | ✓ | 11.6 M | 5 | 0.15 | 47.5 |
| Cosine | Min-SNR | Mean | ✓ | 11.6 M | 5 | 0.15 | 48.2 |
| Inverted Cosine | MNP | Max | ✓ | 11.6 M | 5 | 0.15 | 47.8 |
| Inverted Cosine | MNP | Mean + Max | ✓ | 11.6 M | 5 | 0.15 | 47.4 |
| Inverted Cosine | MNP | Mean | ✗ | 11.6 M | 5 | 0.15 | 44.6 |
| Inverted Cosine | MNP | Mean | ✓ | 3.7 M | 5 | 0.15 | 47.5 |
| Inverted Cosine | MNP | Mean | ✓ | 40.2 M | 5 | 0.15 | **50.6** |
| Inverted Cosine | MNP | Mean | ✓ | 11.6 M | 1 | 0.15 | 45.9 |
| Inverted Cosine | MNP | Mean | ✓ | 11.6 M | 2 | 0.15 | 47.4 |
| Inverted Cosine | MNP | Mean | ✓ | 11.6 M | 5 | 0.00 | 47.2 |
| Inverted Cosine | MNP | Mean | ✓ | 11.6 M | 5 | 0.50 | 47.7 |

# 5 ABLATION STUDIES

In this section, we present the various ablation studies conducted to investigate the effects that different hyperparameter settings have on the diffusion pretraining process and downstream gait recognition tasks. For all experiments, we used the backbone of SMPLGait w/o 3D as our gait feature extractor and Gait3D as the dataset. During transfer learning, no data augmentation was applied, and the pretrained backbone was finetuned with a lower learning rate ($0.1\times$) than the untrained layers. Aside from the hyperparameter being investigated, all other hyperparameters were assigned based on the default pretraining and transfer learning settings. The results of the various ablation studies are summarized in Table 3.

**Noise Scheduler and Loss Weighting**: We explored two kinds of schedulers—a typical cosine scheduler (Nichol & Dhariwal, 2021) and an inverted cosine scheduler (Hudson et al., 2023). For the cosine scheduler, we attempted a uniform weighting strategy, where losses from different timesteps are weighed equally, and the Min-SNR strategy (Hang et al., 2023). As for the inverted cosine scheduler, we attempted uniform weighting and our proposed medium noise prioritization (MNP) weighting strategy, which downweighs losses from both low and high timesteps. We see that focusing on medium-level noise through the combined use of an inverted cosine scheduler and medium noise prioritization weighting strategy worked the best.

**Feature Pooling Method**: We investigated different methods to pool the two-dimensional output of the gait feature extractor into a one-dimensional condition—mean pooling, max pooling, and the sum of the mean and max. Out of the three pooling methods, mean pooling worked the best.

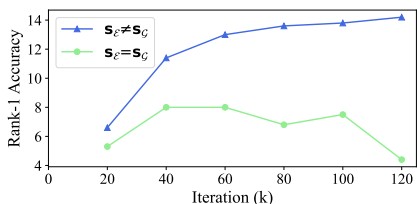

Figure 5: Rank-1 accuracy curves during diffusion pretraining on Gait3D when $s_\mathcal{E} \neq s_\mathcal{G}$ and $s_\mathcal{E} = s_\mathcal{G}$ (SMPLGait w/o 3D).

**Inputs to Denoiser and Gait Feature Extractor**: We investigated what would happen during diffusion pretraining should the input to the gait extractor $s_\mathcal{G}$ and the input to the autoencoder $s_\mathcal{E}$ be identical. Fig. 5 shows the rank-1 gait recognition accuracy curves of the gait feature extractor during the diffusion pretraining process when $s_\mathcal{E} \neq s_\mathcal{G}$ and $s_\mathcal{E} = s_\mathcal{G}$.

When $s_\mathcal{E} \neq s_\mathcal{G}$, we see a steady improvement in gait recognition accuracy during the diffusion pretraining. Yet, when $s_\mathcal{E} = s_\mathcal{G}$, the gait recognition accuracy fluctuates over time, suggesting that what the gait feature extractor is learning during diffusion pretraining is unlikely to be effective gait features. Rather, it could be extracting video information to aid the denoiser in the denoising task.

**Data Augmentation During Pretraining**: We turned off data augmentation during diffusion pretraining to investigate the impact on downstream gait recognition performance. Without any data augmentation during pretraining, a significant drop in the downstream gait recognition performance is observed, highlighting the necessity of data augmentation during diffusion pretraining.

**Size of Denoiser**: We varied the size of the denoiser that is used during the diffusion pretraining process by changing its initial channel dimension. We see that the larger the denoiser, the better the downstream gait recognition performance. This suggests that further improvements can possibly be made to the downstream task by increasing the denoiser size during diffusion pretraining. That said, it would come at the cost of larger memory consumption and longer pretraining time.

**Learning Rate of Denoiser and Gait Feature Extractor**: We investigated how different relative ratios of the learning rate between the gait feature extractor and denoiser, $\frac{l_\mathcal{G}}{l_\mathcal{D}}$, affect the downstream performance. We kept the denoiser learning rate constant at $1e^{-4}$ and varied the learning rate of the gait extractor. We observe that increasing $\frac{l_\mathcal{G}}{l_\mathcal{D}}$ leads to better downstream performance.

**Unconditional Training Probability**: To investigate the effects of varying the unconditional training probability, $P_{\text{uncond}}$, during diffusion pretraining, we explored $P_{\text{uncond}} \in \{0, 0.15, 0.5\}$. We see that our proposed diffusion pretraining approach benefits from classifier-free guidance. However, too high an unconditional training probability ended up hurting the downstream gait recognition performance, likely as the gait feature extractor got updated less. A moderate unconditional training probability yielded the best result.

# 6 CONCLUSION

In summary, we introduce a diffusion pretraining approach for gait recognition in the wild. By simply conditioning a diffusion denoiser with the output of a gait feature extractor, we can pretrain the gait feature extractor to extract relevant features for gait recognition. Initializing the gait recognition model with the pretrained backbone and training it on the downstream gait recognition task further allows us to surpass the performance of its supervised learning counterpart within a much shorter supervised training duration. Our experiments on the Gait3D and GREW datasets demonstrated the broad applicability of our method, achieving rank-1 gait recognition accuracy improvements of up to 7.9% for Gait3D and 4.2% for GREW. We hope our work will spur further interest among researchers in employing diffusion models for representation learning, not only in the gait recognition field but also in other fields as well.

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

# A APPENDIX

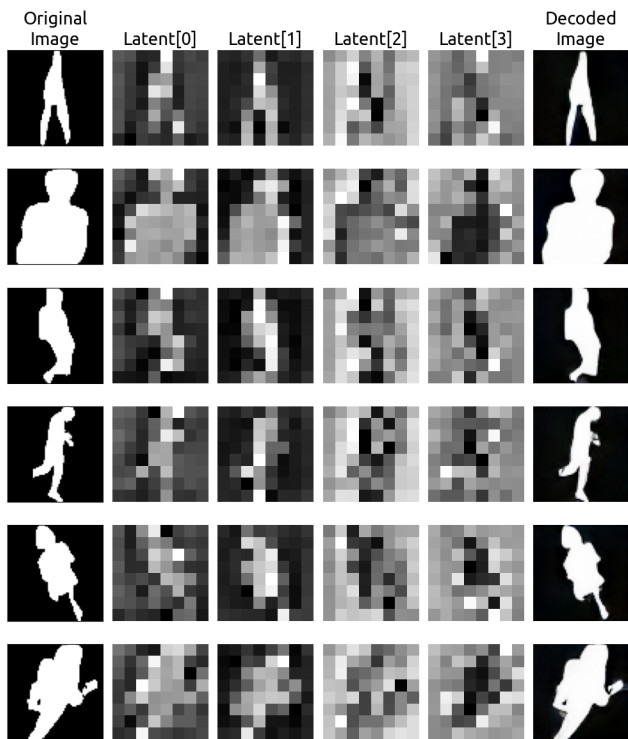

Figure 6: Silhouette reconstruction ability of TAESD on Gait3D and GREW.

## A.1 SILHOUETTE RECONSTRUCTION ABILITY OF TAESD

To ensure that our chosen autoencoder, TAESD, is compatible with grayscale inputs, a small-scale study was conducted, where we encoded 10,000 silhouette frames each from Gait3D and GREW into latent representations and decoded them back to the image space. Based on our results, TAESD was able to encode and decode grayscale images without much loss in detail (Fig. 6), with an average pixel value loss of around 3.8% on Gait3D and 3.4% on GREW.

## A.2 VISUALIZATION OF APPLIED AUGMENTATION

Fig. 7 shows an example of the applied augmentation techniques. Note that the actual input seen by the model will be a compose of these augmentation techniques.

## A.3 REPRODUCTION OF BASELINE RESULTS

Using the OpenGait framework, we reproduced five existing deep gait recognition models—GaitGL, GaitPart, GaitSet, SMPLGait w/o 3D, and GaitBase—both with and without data augmentation during training. Each model was trained using the standard supervised learning objective and evaluated

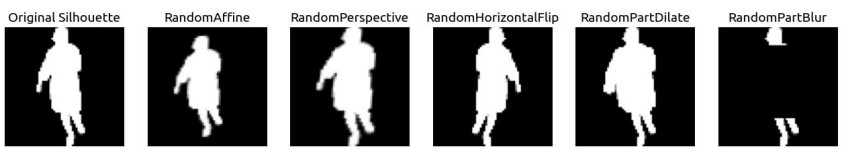

Figure 7: Example of each augmentation technique applied.

Table 4: Reproduced baseline results on Gait3D. mAP and mINP denote mean average precision and mean inverse negative penalty, respectively.

| Model | Gait3D | | | | | |
|---|---|---|---|---|---|---|
| | ✗ Data Aug. | | | ✓ Data Aug. | | |
| | Rank-1 | mAP | mINP | Rank-1 | mAP | mINP |
| **Euclidean Distance** | | | | | | |
| GaitGL | 28.1 | 20.4 | 12.2 | 31.5 | 22.4 | 13.2 |
| GaitPart | 26.6 | 20.1 | 11.8 | 29.7 | 22.0 | 12.8 |
| GaitSet | 37.9 | 30.4 | 17.9 | 39.7 | 32.0 | 19.0 |
| SMPLGait w/o 3D | 44.8 | 35.7 | 21.6 | 42.9 | 33.4 | 19.5 |
| GaitBase | 55.5 | 46.6 | 30.1 | 63.2 | 53.2 | 35.1 |
| **Cosine Distance** | | | | | | |
| GaitGL | 29.2 | 21.1 | 12.3 | 32.4 | 22.9 | 13.7 |
| GaitPart | 31.2 | 23.2 | 13.5 | 38.7 | 29.4 | 17.6 |
| GaitSet | 42.2 | 33.4 | 19.4 | 47.8 | 39.4 | 24.4 |
| SMPLGait w/o 3D | 45.5 | 36.8 | 21.7 | 42.9 | 33.7 | 20.0 |
| GaitBase | 56.5 | 47.3 | 29.1 | 65.8 | 55.8 | 36.5 |

Table 5: Reproduced baseline results on GREW.

| Model | GREW | | | | | |
|---|---|---|---|---|---|---|
| | ✗ Data Aug. | | | ✓ Data Aug. | | |
| | Rank-1 | Rank-5 | Rank-10 | Rank-1 | Rank-5 | Rank-10 |
| **Euclidean Distance** | | | | | | |
| GaitGL | 50.8 | 66.7 | 72.5 | 56.3 | 71.2 | 76.0 |
| GaitSet | 47.5 | 64.7 | 71.2 | 51.7 | 68.1 | 74.0 |
| SMPLGait w/o 3D | 45.8 | 63.3 | 70.0 | 50.9 | 68.6 | 74.9 |
| GaitBase | 58.9 | 73.7 | 79.0 | 59.4 | 74.3 | 79.9 |
| **Cosine Distance** | | | | | | |
| GaitGL | 54.0 | 68.8 | 73.9 | 58.4 | 72.3 | 77.2 |
| GaitSet | 48.1 | 65.2 | 71.4 | 53.1 | 69.7 | 75.7 |
| SMPLGait w/o 3D | 47.6 | 64.8 | 71.2 | 52.1 | 69.0 | 74.8 |
| GaitBase | 58.1 | 73.5 | 78.6 | 61.8 | 76.7 | 81.7 |

for its gait recognition performance. Table 4 and Table 5 present the reproduced baseline results on the Gait3D and GREW datasets, respectively. Since we primarily used a single GPU to train each model, rather than the multi-GPU settings used in OpenGait, our reproduced Euclidean baseline results differ slightly from those reported in the OpenGait repository. Nonetheless, the results are quite similar.

Based on our results, we see that training with cosine distance and Euclidean distance yielded significantly different gait recognition performances. Specifically, cosine distance consistently outperformed Euclidean distance on both the Gait3D and GREW datasets. Consequently, we opted to use cosine distance for both training and evaluation throughout our study. As training for GaitPart is unstable on the GREW dataset when cosine distance is used, we decided to exclude GaitPart from our applicability study on GREW.

### A.4 HYPERPARAMETER SETTINGS

The default hyperparameters used during diffusion pretraining are listed in Table 6. Depending on the memory requirement, we used either RTX 3090 or RTX A6000 GPUs for pretraining, as indicated in Table 7. To ensure fairness, both the reproduction of the baseline results in Appendix A.3 and our transfer learning experiments were conducted using the same GPU shown in Table 8. For GaitBase with data augmentation, as it was impossible to achieve similar performance as stated in OpenGait using a single GPU, we had to turn to use the multi-GPU settings originally used by the authors.

### A.5 COMPLETE PRETRAINING FINDINGS

Fig. 8 and Fig. 9 show the gait recognition performance for the different gait feature extractors used at different checkpoints of the pretraining process with the Gait3D and GREW datasets, respectively. For all cases, we observe a steady improvement in gait recognition performance during the diffusion pretraining process.

Table 6: Default hyperparameters used during diffusion pretraining.

| Hyperparameter | Value |
|---|---|
| **Latent Diffusion Settings** | |
| Autoencoder | Tiny AutoEncoder for Stable Diffusion |
| Diffusion Timestep | 1000 |
| Noise Scheduler | Inverted Cosine |
| Loss-Weighting Strategy | Medium-Level Noise Prioritization |
| Condition Pooling Method | Mean |
| **Video Diffusion Model Settings** | |
| Initial Kernel Size | 5 |
| Initial Dimension | 64 |
| Dimension Multiplier | [1, 2, 4] |
| Number of Attention Heads | 8 |
| Attention Dimension (Per Head) | 32 |
| Number of Input Frames | 30 |
| Timestep Condition Dimension | 256 |
| GroupNorm Number of Groups | 32 |
| **Pretraining Settings** | |
| Batch Size | 64 (Gait3D), 128 (GREW) |
| Optimizer | AdamW |
| Learning Rate (Denoiser) | $1e^{-4}$ |
| Learning Rate (Gait Feature Extractor) | $5e^{-4}$ |
| Learning Rate Scheduler | CosineAnnealingLR |
| Training Iterations | 120000 |
| Warmup Iterations | 2000 |
| Warmup Start Factor | 0.01 |
| **Input Data Augmentation Settings** | |
| RandomAffine Probability | 0.2 |
| RandomPerspective Probability | 0.2 |
| RandomHorizontalFlip Probability | 0.5 |
| RandomPartDilate Probability | 0.2 |

Table 7: GPU used during diffusion pretraining.

| Method | Gait3D | GREW |
|---|---|---|
| GaitGL | RTX 3090 | RTX A6000 |
| GaitPart | RTX 3090 | - |
| GaitSet | RTX 3090 | RTX 3090 |
| SMPLGait w/o 3D | RTX 3090 | RTX 3090 |
| GaitBase | RTX 3090 | RTX A6000 |

Table 8: GPU used during reproduction of baseline and transfer learning.

| Method | Gait3D | | GREW | |
|---|---|---|---|---|
| | ✗ Data Aug. | ✓ Data Aug. | ✗ Data Aug. | ✓ Data Aug. |
| GaitGL | RTX 3090 | RTX 3090 | RTX A6000 | RTX A6000 |
| GaitPart | RTX 3090 | RTX 3090 | - | - |
| GaitSet | RTX 3090 | RTX 3090 | RTX 3090 | RTX 3090 |
| SMPLGait w/o 3D | RTX 3090 | RTX 3090 | RTX 3090 | RTX 3090 |
| GaitBase | RTX 3090 | RTX A6000 | RTX A6000 | $4 \times$ A100 |

Fig. 10 and Fig. 11 show the recorded mean cosine distance of the gait features of anchor-positive pairs and the gait features of anchor-negative pairs within a batch during pretraining on the Gait3D and GREW datasets, respectively, for the different gait feature extractors used. We see that regardless of the gait feature extractor used, the difference in the cosine distances between the anchor-positive pairs and anchor-negative pairs increases and stabilises during diffusion pretraining.

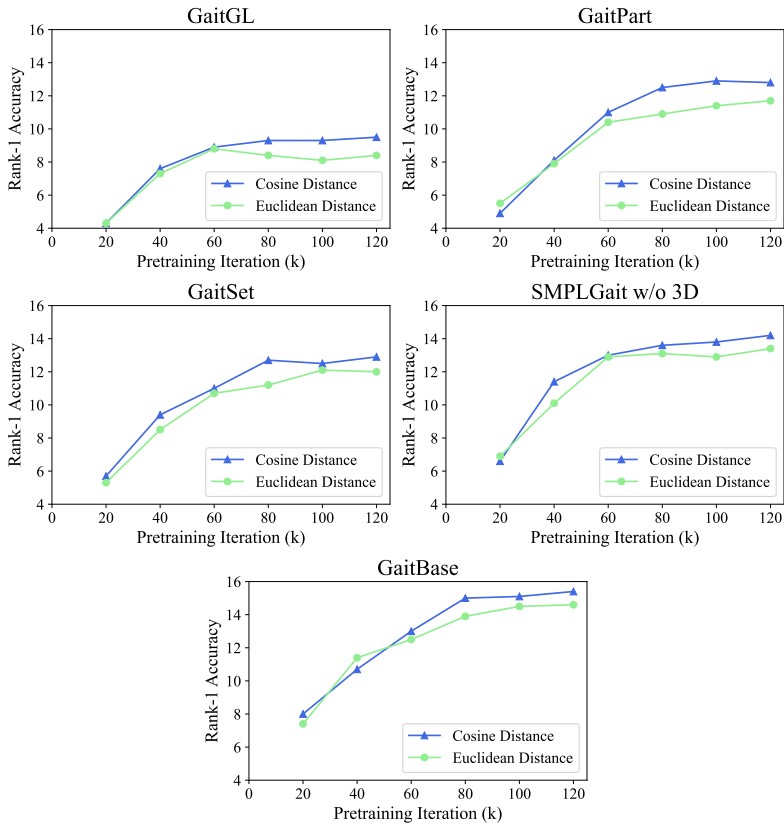

Figure 8: Rank-1 accuracy curves during diffusion pretraining on Gait3D.

### A.6    TRANSFER LEARNING RESULTS WITH DIFFERENT $r$ VALUES

Table 9 shows the rank-1 accuracy of the various models on the Gait3D and GREW datasets with different ratios of the learning rate of the pretrained layers to that of the untrained layers, $r$, attempted.

### A.7    CROSS-DOMAIN EVALUATION

With the checkpoints obtained from training, we evaluated each of its cross-domain performance on three other datasets (Table 10). For the checkpoints trained on Gait3D, we evaluated on CASIA-B, OU-MVLP, and GREW. As for those trained on GREW, we evaluated on CASIA-B, OU-MVLP, and Gait3D. For all evaluation, we used cosine distance as the distance metric.

As seen from Table 10, our proposed method leads to improved cross-domain performance for the majority of the methods, demonstrating that the inclusion of diffusion pretraining allows the methods to converge to a better solution not just on the dataset it is trained on but also on other datasets.

### A.8    WEIGHT DISTRIBUTION ANALYSIS

The default method of weight initialization used by the OpenGait framework is the Xavier uniform initialization (Glorot & Bengio, 2010). We visualize how the distribution of the weights of the final

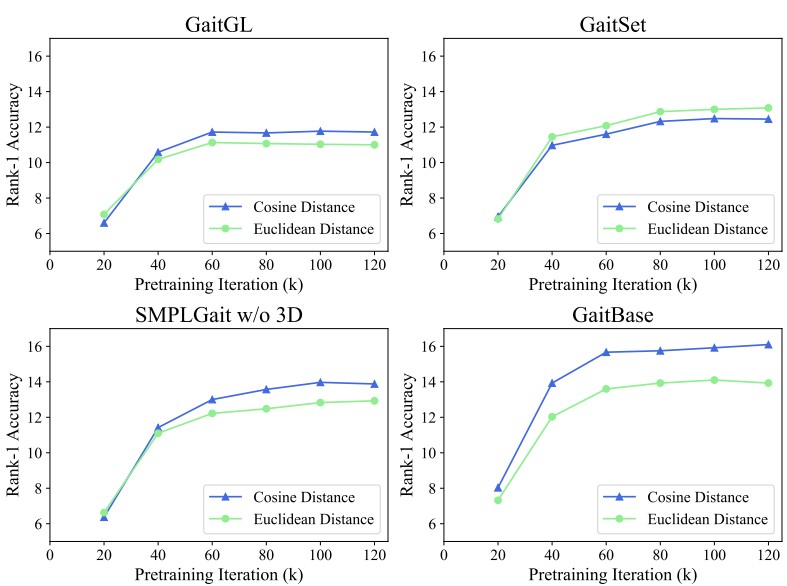

Figure 9: Rank-1 accuracy curves during diffusion pretraining on GREW.

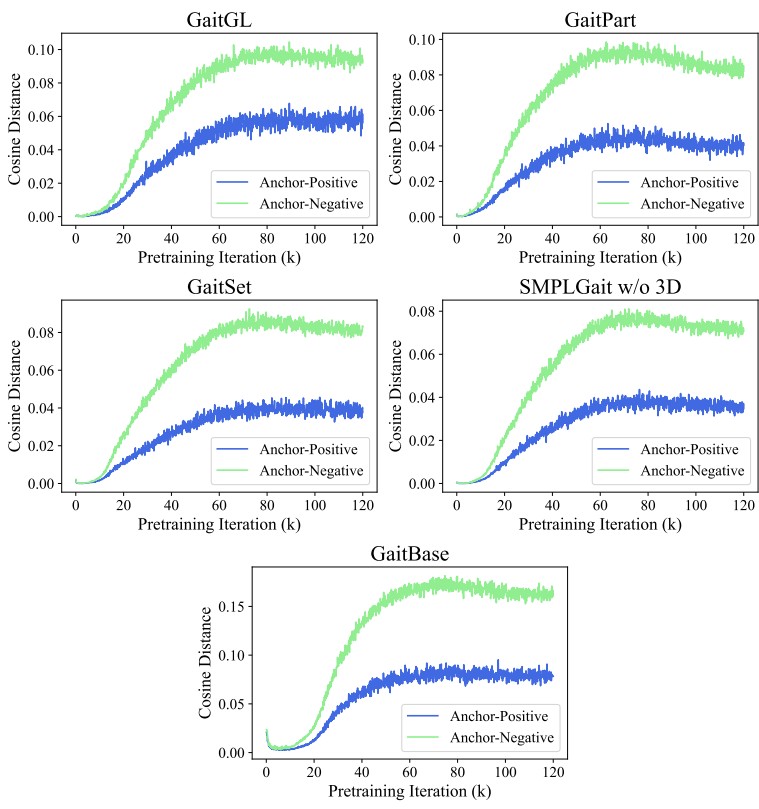

Figure 10: Mean cosine distance of anchor-positive pair and anchor-negative pair during diffusion pretraining on Gait3D.

convolution layer in one of the models, SMPLGait w/o 3D, changes as it is pretrained by diffusion followed by transfer learning (Fig. 12). With diffusion pretraining followed by transfer learning, the

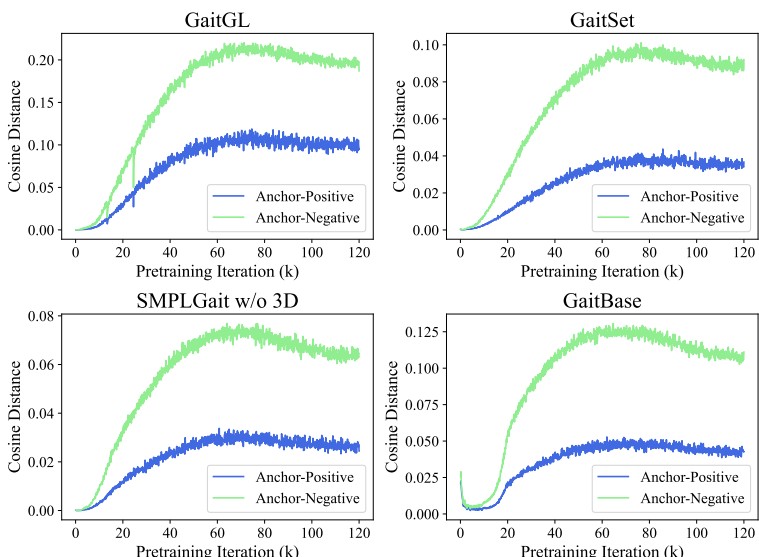

Figure 11: Mean cosine distance of anchor-positive pair and anchor-negative pair during diffusion pretraining on GREW.

Table 9: Rank-1 accuracy on the Gait3D and GREW datasets for different values of $r$ attempted during transfer learning. Values in bold denote the best results for each model on Gait3D and GREW.

| Method | Gait3D | | | GREW | | |
|---|---|---|---|---|---|---|
| | Rank-1 Accuracy (%) | | Train Iter. | Rank-1 Accuracy (%) | | Train Iter. |
| | ✗ Data Aug. | ✓ Data Aug. | $(\times 10^3)$ | ✗ Data Aug. | ✓ Data Aug. | $(\times 10^3)$ |
| | | $r = 0.1$ | | | $r = 0.1$ | |
| GaitGL | **34.4** | **34.4** | 120 + 60 | 46.2 | 48.0 | 120 + 125 |
| GaitPart | 32.5 | 39.1 | 120 + 60 | - | - | - |
| GaitSet | 40.1 | 47.3 | 120 + 60 | 45.1 | 46.4 | 120 + 125 |
| SMPLGait w/o 3D | 49.1 | 57.2 | 120 + 60 | 43.8 | 44.5 | 120 + 125 |
| GaitBase | 47.5 | 55.8 | 120 + 60 | 50.7 | 52.7 | 120 + 90/120 |
| | | $r = 0.5$ | | | $r = 0.5$ | |
| GaitGL | 33.1 | 32.3 | 120 + 60 | 54.3 | 56.5 | 120 + 125 |
| GaitPart | **35.7** | **41.7** | 120 + 60 | - | - | - |
| GaitSet | **45.0** | **49.9** | 120 + 60 | 50.9 | 53.3 | 120 + 125 |
| SMPLGait w/o 3D | **53.4** | **60.7** | 120 + 60 | 49.3 | 51.2 | 120 + 125 |
| GaitBase | 60.2 | 68.2 | 120 + 60 | **58.5** | **62.0** | 120 + 90/120 |
| | | $r = 1.0$ | | | $r = 1.0$ | |
| GaitGL | 29.4 | 28.6 | 120 + 60 | **56.3** | **58.6** | 120 + 125 |
| GaitPart | 33.0 | 38.4 | 120 + 60 | - | - | - |
| GaitSet | 43.4 | 48.5 | 120 + 60 | **52.0** | **55.4** | 120 + 125 |
| SMPLGait w/o 3D | 51.1 | 58.6 | 120 + 60 | **51.8** | **54.1** | 120 + 125 |
| GaitBase | **62.3** | **69.7** | 120 + 60 | 58.5 | 61.6 | 120 + 90/120 |

model converges to a new solution with a different weight distribution, which may explain the better performance of our proposed approach than the baselines with only supervised learning.

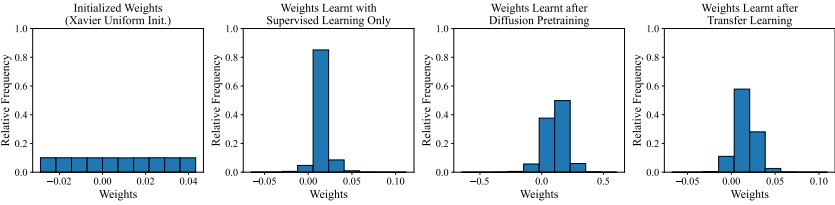

Figure 12: Weight distribution of the final convolution layer of SMPLGait w/o 3D.

Table 10: Cross-domain rank-1 accuracy evaluation results attained with the checkpoints trained on Gait3D and GREW.

| Source Dataset | Data Aug. | Method | Target Dataset | | | | | |
|---|---|---|---|---|---|---|---|---|
| | | | CASIA-B | | | OUMVLP | Gait3D | GREW |
| | | | NM | BG | CL | | | |
| Gait3D | ✗ | **Reproduced Baseline** | | | | | | |
| | | GaitGL | 59.0 | 48.1 | 17.3 | 22.7 | - | 10.5 |
| | | GaitPart | 56.8 | 44.2 | 15.5 | 16.9 | - | 12.1 |
| | | GaitSet | 62.0 | 45.3 | 11.9 | 19.9 | - | 15.0 |
| | | SMPLGait w/o 3D | 66.9 | 51.8 | 13.7 | 24.4 | - | 17.5 |
| | | GaitBase | 72.8 | 61.5 | 18.9 | 32.2 | - | 21.0 |
| | | **Diffusion Pretraining + Transfer Learning** | | | | | | |
| | | GaitGL | 64.7 (↑ 5.7) | 52.7 (↑ 4.1) | 19.0 (↑ 1.7) | 25.2 (↑ 2.5) | - | 13.8 (↑ 3.3) |
| | | GaitPart | 61.3 (↑ 4.5) | 46.5 (↑ 2.3) | 14.6 (↓ 0.9) | 20.1 (↑ 3.2) | - | 13.8 (↑ 1.7) |
| | | GaitSet | 63.4 (↑ 1.4) | 46.5 (↑ 1.2) | 12.6 (↑ 0.7) | 20.9 (↑ 1.0) | - | 15.9 (↑ 0.9) |
| | | SMPLGait w/o 3D | 71.4 (↑ 4.5) | 55.0 (↑ 3.2) | 16.2 (↑ 2.5) | 28.2 (↑ 3.8) | - | 20.7 (↑ 3.2) |
| | | GaitBase | 74.2 (↑ 1.4) | 62.3 (↑ 0.8) | 19.9 (↑ 1.0) | 35.4 (↑ 3.2) | - | 22.7 (↑ 1.7) |
| | ✓ | **Reproduced Baseline** | | | | | | |
| | | GaitGL | 68.1 | 55.2 | 17.1 | 26.5 | - | 11.4 |
| | | GaitPart | 59.8 | 43.0 | 13.8 | 22.4 | - | 13.2 |
| | | GaitSet | 65.8 | 48.7 | 11.8 | 24.7 | - | 18.3 |
| | | SMPLGait w/o 3D | 69.8 | 53.8 | 12.9 | 25.7 | - | 17.5 |
| | | GaitBase | 78.7 | 65.9 | 16.4 | 39.3 | - | 25.2 |
| | | **Diffusion Pretraining + Transfer Learning** | | | | | | |
| | | GaitGL | 70.9 (↑ 2.8) | 60.2 (↑ 5.0) | 18.4 (↑ 1.3) | 29.2 (↑ 2.7) | - | 14.2 (↑ 2.8) |
| | | GaitPart | 67.7 (↑ 7.9) | 53.4 (↑ 10.4) | 15.0 (↑ 1.2) | 23.7 (↑ 1.3) | - | 15.8 (↑ 2.6) |
| | | GaitSet | 68.4 (↑ 2.6) | 49.2 (↑ 2.7) | 11.8 (-) | 24.8 (↑ 0.1) | - | 18.0 (↓ 0.3) |
| | | SMPLGait w/o 3D | 75.4 (↑ 5.6) | 60.5 (↑ 6.7) | 15.5 (↑ 2.6) | 33.7 (↑ 8.0) | - | 23.2 (↑ 5.7) |
| | | GaitBase | 78.8 (↑ 0.1) | 66.4 (↑ 0.5) | 16.2 (↓ 0.2) | 39.4 (↑ 0.1) | - | 25.2 (-) |
| GREW | ✗ | **Reproduced Baseline** | | | | | | |
| | | GaitGL | 69.9 | 57.0 | 29.7 | 25.9 | 21.3 | - |
| | | GaitSet | 62.8 | 46.2 | 20.6 | 19.0 | 18.0 | - |
| | | SMPLGait w/o 3D | 66.6 | 53.6 | 22.8 | 21.4 | 18.9 | - |
| | | GaitBase | 67.8 | 54.9 | 24.3 | 25.2 | 22.2 | - |
| | | **Diffusion Pretraining + Transfer Learning** | | | | | | |
| | | GaitGL | 70.1 (↑ 0.2) | 57.1 (↑ 0.1) | 30.5 (↑ 0.8) | 28.2 (↑ 2.3) | 23.0 (↑ 1.7) | - |
| | | GaitSet | 64.9 (↑ 2.1) | 48.7 (↑ 2.5) | 22.3 (↑ 1.7) | 20.5 (↑ 1.5) | 19.6 (↑ 1.6) | - |
| | | SMPLGait w/o 3D | 68.3 (↑ 1.7) | 54.7 (↑ 1.1) | 24.1 (↑ 1.3) | 24.0 (↑ 2.6) | 22.9 (↑ 4.0) | - |
| | | GaitBase | 69.8 (↑ 2.0) | 57.7 (↑ 2.8) | 26.4 (↑ 2.1) | 26.6 (↑ 1.4) | 23.6 (↑ 1.4) | - |
| | ✓ | **Reproduced Baseline** | | | | | | |
| | | GaitGL | 73.4 | 61.3 | 31.1 | 30.2 | 26.0 | - |
| | | GaitSet | 66.4 | 52.1 | 23.1 | 22.6 | 22.4 | - |
| | | SMPLGait w/o 3D | 70.7 | 55.8 | 23.6 | 26.5 | 23.4 | - |
| | | GaitBase | 70.3 | 58.7 | 25.2 | 29.9 | 28.0 | - |
| | | **Diffusion Pretraining + Transfer Learning** | | | | | | |
| | | GaitGL | 71.6 (↓ 1.8) | 61.4 (↑ 0.1) | 33.4 (↑ 2.3) | 30.7 (↑ 0.5) | 27.2 (↑ 1.2) | - |
| | | GaitSet | 65.8 (↓ 0.6) | 50.0 (↓ 2.1) | 21.7 (↓ 1.4) | 23.4 (↑ 0.8) | 22.6 (↑ 0.2) | - |
| | | SMPLGait w/o 3D | 70.5 (↓ 0.2) | 54.7 (↓ 1.1) | 23.3 (↓ 0.3) | 26.7 (↑ 0.2) | 26.8 (↑ 3.4) | - |
| | | GaitBase | 71.9 (↑ 1.6) | 59.7 (↑ 1.0) | 24.6 (↓ 0.6) | 30.2 (↑ 0.3) | 28.6 (↑ 0.6) | - |

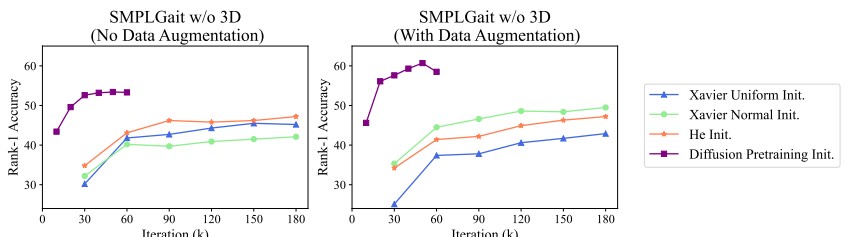

Figure 13: Rank-1 accuracy on Gait3D for SMPLGait w/o 3D with different initialization methods.

## A.9 OTHER WEIGHT INITIALIZATION METHODS

Other than the default Xavier uniform initialization, we attempted other initialization including Xavier normal initialization and He initialization (He et al., 2015). Fig. 13 shows a comparison of the final Rank-1 gait recognition accuracy on Gait3D for the various initialization methods attempted on SMPLGait w/o 3D. As seen in Fig. 13, our diffusion pretrained checkpoint allows the model to achieve higher gait recognition accuracy within a shorter number of supervised training iterations.

