# OpenReview forum: "Diffusion Pretraining for Gait Recognition in the Wild"
_ICLR.cc/2025/Conference — ICLR 2025 Conference Withdrawn Submission_

### Official Review · Reviewer_CXTE · 2024-10-29

**Soundness:** 2
**Presentation:** 3
**Contribution:** 3
**Rating:** 5
**Confidence:** 5

**Summary:**

This research makes a good attempt to use diffusion-based representation learning on gait recognition by using gait models’ output as a condition for a latent diffusion model.

**Strengths:**

1. Good Novelty. The paper explores diffusion methods on gait recognition task, and reveals an interesting and useful phenomenon.

2. Good Improvement. Abundant experiments show that diffusion pertaining considerable improves the existing gait methods.

**Weaknesses:**

1. Question on Pre-training. The pre-training process seems a knowledge distillation between diffusion models and gait models. The experimental improvement shows diffusion models are good at identifying human in some case. Therefore, the authors are expected to directly fine-tune the TAESD (your diffusion encoder) on the downstream gait recognition task. If we can get higher performance by fine-tuning diffusion itself, why we need to pre-train other gait methods by diffusion models?

2. The reduction of supervised training iterations seems weak, since you have to pretrain other gait models at first. Tab.1 also shows that the number of original training iteration is  almost equal to that of pertaining + transfer learning.

3. The authors are suggested to try some stronger gait models (like deepgaitv2) as the backbone.

**Questions:**

If the authors can address my concerns, I would like to improve the final rating.

---

> ### Author Response · Authors · 2024-11-19
> **Official Response to Reviewer CXTE (Part 1 of 2)**
>
> Dear Reviewer,
>
> Thank you for taking the time to review our paper. We are happy to hear that you found our work novel and interesting. We are also glad that found our approach effective in generating considerable improvement for existing gait methods.
>
> The following are our responses to your comments on the shortcomings of our paper.
>
> **Comment 1**: *Question on Pre-training. The pre-training process seems a knowledge distillation between diffusion models and gait models. The experimental improvement shows diffusion models are good at identifying human in some case. Therefore, the authors are expected to directly fine-tune the TAESD (your diffusion encoder) on the downstream gait recognition task. If we can get higher performance by fine-tuning diffusion itself, why we need to pre-train other gait methods by diffusion models?*
>
> There might be some misunderstanding regarding the diffusion pretraining process. The pretraining process involves training both the diffusion model and gait recognition model backbone from scratch, where their weights are both initialized randomly at the beginning and are trained simultaneously with the diffusion loss in the backward pass. As such, we would say that there is no knowledge distillation between the diffusion model and the gait recognition model backbone.
>
> As for the second part of the comment, we are not sure if we understood it correctly, but you are suggesting that we should finetune TAESD / the diffusion model on the downstream gait recognition task. However, note that for the downstream task, we are only using the pretrained gait recognition model backbone. The diffusion model and TAESD are not involved in this phase of finetuning. Indeed, it may be possible to include them when finetuning on the downstream task. However, it would not make sense to compare the gait recognition performance with the baselines due to the huge difference in model architecture. Hence, while we appreciate the suggestion, we have to ensure a fair comparison and only use the pretrained gait recognition model backbone for the downstream task. Let us know if we understood your question correctly.
>
> **Comment 2**: *The reduction of supervised training iterations seems weak, since you have to pretrain other gait models at first. Tab.1 also shows that the number of original training iteration is almost equal to that of pertaining + transfer learning.*
>
> We understand your concern regarding the training iteration count of our proposed approach. If we can refer you to Figure 4 in our paper, we show that after 120,000 training iterations on Gait3D, the gait recognition models (GaitGL, GaitPart, SMPLGait w/o 3D) only require as little as around 20,000 additional iterations of finetuning on the downstream task to outperform its corresponding baseline. In such cases, the cumulative training iteration (120,000 + 20,000) is less than the training iteration count (180,000) for the baseline. The reason why we made the training iteration for pretraining + transfer learning equal to the original training iteration in Table 1 is to demonstrate the possible accuracy performance gain on the gait recognition task with our approach.
>
> Furthermore, one of the main reasons why the pretraining iteration is long is currently due to the fact that we have to train both the diffusion model and the gait recognition model backbone from scratch. This is because there does not exist a publicly available pretrained diffusion model for gait silhouette generation. Should we have a pretrained diffusion model for gait silhouette generation, we believe that the pretraining iteration can be greatly reduced. This is especially true since the size of the diffusion model is larger than most of the gait recognition model backbones we have experimented with. We hope that you understand the additional overhead that we have to introduce due to the current constraint we have.

---

> ### Author Response · Authors · 2024-11-19
> **Official Response to Reviewer CXTE (Part 2 of 2)**
>
> **Comment 3**: *The authors are suggested to try some stronger gait models (like deepgaitv2) as the backbone.*
>
> Thank you for the suggestion. As you have suggested, we have previously tried out other recently released models such as DeepGaitV2 and SwinGait that are available in the OpenGait repository. However, we found it difficult to replicate the baseline accuracy performance of these models. Take DeepGaitV2 for instance, even though the authors managed to achieve a Rank 1 gait recognition accuracy of 74.4% on Gait3D, we only managed to achieve a baseline accuracy of only 42.3% with the same configuration settings. Because of such a huge discrepancy, we decided not to proceed with them for our study as it would no longer be a fair comparison. Moreover, these models are much deeper and are much more computationally demanding to experiment with different hyperparameters, compared to the existing models we used.
>
> We hope that we have addressed all, if not most, of your current concerns. Nonetheless, if you still have any other doubts about our paper or response, feel free to reach out to us and we would be more than happy to clarify them. If all your concerns have been addressed, we would greatly appreciate it if you could reconsider the rating of our paper.

---

> ### Comment · Reviewer_CXTE · 2024-11-25
>
> I have tested DeepGaitV2 and SwinGait using OpenGait and achieved performance comparable to the reported results. Additionally, I noticed that OpenGait has released pre-trained checkpoint files on Huggingface (https://huggingface.co/opengait/OpenGait/), which allow for result reproduction without the need for training.
>
> Since I consider that employing state-of-the-art models as baselines is essential, this rebuttal might not be sufficient to significantly improve my rating. The submission represents one of the first attempts at diffusion-based gait recognition, which is commendable, but further improvements are needed to meet the standards of a top-tier conference.

---

> > ### Author Response · Authors · 2024-11-25
> > **Official Response to Reviewer CXTE**
> >
> > Dear Reviewer,
> >
> > We appreciate your acknowledgement of our work as one of the first attempts at diffusion-based gait recognition. We also recognize the importance of experimenting with state-of-the-art models and understand how this influences your evaluation. While we might be unable to change your decision, we would still like to address your concern below.
> >
> > As you have done, we have also previously tested DeepGaitV2 and SwinGait using OpenGait’s provided checkpoint and indeed, their performance is as stated in the OpenGait repository. However, the problem of reproducibility comes into the picture when we try to train them from scratch with the provided script.
> >
> > The main reason why we had to reproduce their results is to ensure fairness. If we are unable to guarantee that the baseline is capable of reaching the performance as stated, we will not be certain of our approach’s effectiveness.
> >
> > Moreover, we found that the usage of different GPUs and the number of GPUs used have some influence on the final accuracy performance achieved by the model. As such, in our experiments, we made it a point to ensure that our approach uses the same GPUs during training and evaluation as the baseline for fair comparison.
> >
> > All in all, as we would like to ensure that our work is fair and reproducible, we decided not to include DeepGaitV2 and SwinGait. While it might be possible for us to tinker with different hyperparameter settings and eventually arrive at their performance, it is extremely expensive for us especially since state-of-the-art models have much deeper architectures and the hyperparameter space is huge.
> >
> > Even without the inclusion of deep state-of-the-art models, we hope that you can view our work as a way to improve existing methods that are more suited to resource-constrained environments where heavy reliance on deep state-of-the-art models may not be practical.
> >
> > Thank you for your constructive feedback and for recognizing the novelty of our contribution.

---

### Official Review · Reviewer_X1wE · 2024-11-01

**Soundness:** 3
**Presentation:** 4
**Contribution:** 2
**Rating:** 5
**Confidence:** 4

**Summary:**

The authors try to make use of the superiority of the diffusion model in gait recognition, serving as a pretraining method. Based on the experiments, the proposed pre-training method have the ability to recognize the people alone. When it is trainable in the downstream task, the performance is higher than the models without pertaining, demonstrating the effectiveness of the proposed method.

**Strengths:**

1. The work is a good attempt to use the diffusion model in gait recognition serving as a pre-training model.
2.  The authors did extensive experiments to demonstrate the details and effectiveness of the model.
3. The present is clear and it is easy to understand.

**Weaknesses:**

1. The motivation is not that clear. What feature of the diffusion model is potentially helpful for gait recognition
2. What does 'focus on gait information rather than video information' mean in sec 3.1.5. What is the video information defined here?
3. The sampling method is not clear. Sequential gait recognitions, such as GaitGL, still use consecutive 30-frames. So what is the difference between these two inputs?
4. The whole paper tells what the authors did but with little analysis, such as why the diffusion model could be helpful and why the structure is like this.

**Questions:**

The figure could be improved. If the different frames are selected, the figure should have this kind of information rather than the same silhouettes.
The experiments section only demonstrates the tables' content but lacks analysis, such as why there is improvement and why some works do not work well.

---

> ### Author Response · Authors · 2024-11-19
> **Official Response to Reviewer X1wE (Part 1 of 2)**
>
> Dear Reviewer,
>
> Thank you for taking the time to review our paper. We are happy to hear that you found our paper clear and easy to understand. We are also glad that you appreciated the extensive experiments and ablations studies that we conducted to demonstrate the effectiveness of our approach.
>
> The following are our responses to your comments on the shortcomings of our paper as well as the questions posed.
>
> **Comment 1**: *The motivation is not that clear. What feature of the diffusion model is potentially helpful for gait recognition?*
>
> We believe that there might be some misunderstanding here. Our paper focuses on using diffusion pretraining to pretrain the backbone of a gait recognition model. The features provided by the gait recognition model backbone are used as a condition for the diffusion model to reconstruct the inputs. With diffusion pretraining, we are training the gait recognition model backbone to provide the right condition for input reconstruction. We then use this pretrained gait recognition model backbone for the downstream gait recognition task. As such, our approach does not involve the use of the features from the diffusion model for gait recognition.
>
> **Comment 2**: *What does 'focus on gait information rather than video information' mean in sec 3.1.5. What is the video information defined here?*
>
> In order for the diffusion model to reconstruct the input precisely, what the gait recognition model backbone provides as a condition to the diffusion model should be something that is common to both the inputs of the gait recognition model backbone and the diffusion model.
>
> When the inputs to the gait recognition model backbone and the diffusion model are exactly the same, it is highly likely that the gait recognition model backbone converges to a solution where it learns to extract the image features of each frame to help with the generation process, since that should be easier and more direct than learning to extract gait features. A possible example is learning how the background pixels are distributed. However, that is not what we want the gait recognition model backbone to learn.
>
> Instead, what we want it to learn is to extract the gait features that are common to both the inputs of the gait recognition model backbone and the diffusion model. To do so, we made it a point to ensure that the inputs to the gait recognition model and the diffusion model are different.
>
> To prove our point, we kindly refer you to Figure 5 in our ablation studies. We show that when the inputs to the gait recognition model and the diffusion model are the same, the gait recognition accuracy fluctuates over time, suggesting that the gait model recognition backbone may not be extracting the gait features effectively. It might be instead learning to extract the image features of each silhouette frame to help with the reconstruction.
>
> **Comment 3**: *The sampling method is not clear. Sequential gait recognitions, such as GaitGL, still use consecutive 30-frames. So what is the difference between these two inputs?*
>
> Indeed, sequential gait recognition, such as GaitGL and GaitPart, uses consecutive frames for the inputs and our diffusion model also accepts consecutive frames as it is a video diffusion model. While the gait recognition backbone model and diffusion model accept consecutive frames, do note that the gait silhouette sequences are often more than 30 frames. As such, it is highly likely that the sampled 30 frames input to the gait recognition model backbone is different from that of the diffusion model. Just to illustrate, if the sequence length of a gait sequence is 60, the diffusion model might receive the first 30 frames of the gait sequence while the gait recognition model backbone might receive the last 30 frames.
>
> In the event that the sequence length of the gait sequence is less than 30 frames, the sampling algorithm will first repeat the gait sequence until it is at least equal to or greater than 30 frames. For example, if the gait sequence is only 25 frames, it will first be duplicated to 50 frames and 30 consecutive frames will be sampled.
>
> Only in the event that the sequence length of a gait sequence is exactly 30 will the input to the diffusion model and gait recognition model be exactly the same. However, such cases are rare.

---

> ### Author Response · Authors · 2024-11-19
> **Official Response to Reviewer X1wE (Part 2 of 2)**
>
> **Comment 4**: *The whole paper tells what the authors did but with little analysis, such as why the diffusion model could be helpful and why the structure is like this.*
>
> As mentioned in the response to **Comment 1**, the diffusion model is used here to help pretrain the backbone of a gait recognition model. We show that it is possible for the gait recognition model backbone to learn to extract some discriminative features for gait recognition even though it is simply providing the features as a condition for input reconstruction. This can be seen in Figure 2, where the gait recognition performance steadily improves during diffusion pretraining. Perhaps, as gait is unique to each individual, to generate a gait sequence of a particular individual, a unique condition is required, which may enforce the learning of some form of discriminative gait feature in the process.
>
> The architecture is as such because it is based on a common structure of conditional diffusion models, where a diffusion model is conditioned with additional details such as the output of another model.
> - The diffusion encoder (TAESD) is used to provide a lightweight solution to compress our inputs into the latent space to reduce memory and computational demand.
> - A video diffusion model is used as we are dealing with temporal gait sequences. Due to the lack of open-sourced pretrained gait silhouette sequence diffusion model, we adopted an open-sourced video diffusion model.
> - The gait recognition model backbone is the model of interest that we wish to pretrain, and we do so by providing its output as a condition to the video diffusion model.
>
> **Question 1**: *The figure could be improved. If the different frames are selected, the figure should have this kind of information rather than the same silhouettes.*
>
> We have taken into account your feedback, and we have adjusted Figure 1 accordingly.
>
> **Question 2**: *The experiments section only demonstrates the tables' content but lacks analysis, such as why there is improvement and why some works do not work well.*
>
> We believe the reason why there is an improvement in the gait recognition accuracy after diffusion pretraining is due to the fact that diffusion pretraining provides a decent initialization point compared to simply random initialization. We have seen that the gait recognition performance steadily increases during diffusion pretraining, demonstrating that the gait recognition backbone has learnt some discriminative features for gait recognition, which may be beneficial for the downstream task.
>
> As for the second part of the question on why some works do not work as well, most of the hyperparameters that we employed for diffusion pretraining are based on our ablation studies with the SMPLGait w/o 3D model. Perhaps that might explain the poorer performance increase with the GaitBase model. However, it seems to have worked decently well with other models that we have experimented with. If we had the liberty to tune the hyperparameters for each model, we believe the performance gain might be better.
>
> We hope that we have addressed all, if not most, of your current concerns. Nonetheless, if you still have any other doubts about our paper or response, feel free to reach out to us and we would be more than happy to clarify them. If all of your concerns have been addressed, we would greatly appreciate it if you could reconsider the rating of our paper.

---

> ### Comment · Reviewer_X1wE · 2024-11-25
>
> Comment 1: Regarding this question, I share the same concern as Reviewer 8BzL regarding the rationale for applying diffusion in the paper and how the diffusion model enhances the model's discrimination power. Based on the response, it remains unclear. I do not know the motivation, like why it should work. Additionally, while the results demonstrate improvement, there is no clear explanation and proof of the underlying mechanism driving this enhancement.
>
> Comment 2: I am still not clear what is the so-called video information mentioned in the paper.

---

> ### Author Response · Authors · 2024-11-26
> **Official Response to Reviewer X1wE**
>
> Dear Reviewer,
>
> We appreciate your feedback regarding our rebuttal. The following are our responses to your comments.
>
> **Comment 1**: *Regarding this question, I share the same concern as Reviewer 8BzL regarding the rationale for applying diffusion in the paper and how the diffusion model enhances the model's discrimination power. Based on the response, it remains unclear. I do not know the motivation, like why it should work. Additionally, while the results demonstrate improvement, there is no clear explanation and proof of the underlying mechanism driving this enhancement.*
>
> We understand your concern regarding the rationale for applying diffusion in the paper. Allow us to clarify our motivation below.
>
> In recent years, we have seen how conditional diffusion models are able to generate more specific data based on certain conditions provided. One way of conditioning diffusion models is through the use of discriminative features that are provided by a discriminative model. If it is possible to train a diffusion model to output more precise data when conditioned on such discriminative features, can we do the reverse – where we train a discriminative model to provide discriminative features for precise input reconstruction using the same diffusion loss? That is what we are trying to investigate in our paper by applying to a specific application – gait recognition in the wild.
>
> To tackle this research question, we first pretrain the gait recognition model backbone by conditioning its output on a diffusion model and using just the diffusion loss. We then trained both the diffusion-pretrained model and the randomly initialized model on the gait recognition task. If the models have not learnt anything useful during diffusion pretraining, we would expect their performance (despite diffusion pretraining) to be on par with their randomly initialized baselines.  However, based on our experiments, this was not the case. As you can see in Figure 4, methods that were pretrained with diffusion were able to outperform their corresponding baselines within as little as 20,000 training iterations on the downstream gait recognition task as compared to 180,000 training iterations for the baselines when randomly initialized. This demonstrates that the gait recognition model backbone has indeed learnt something useful for gait recognition during diffusion pretraining. In fact, based on the distances in the cosine space in Figure 3, we can observe some discriminative nature of the conditions (gait features) learnt during diffusion pretraining.
>
> As for the theoretical proof of the underlying mechanism driving this phenomenon, we are still unable to provide them at the moment. However, based on our findings, we hypothesize that just like how diffusion models can be trained to generate specific data based on given conditions (e.g. features of a discriminative model), it is also possible to learn the reverse – where a discriminative model is trained with diffusion to provide sufficiently discriminative conditions for diffusion models to generate specific data.
>
> While theoretical proofs can be valuable, it is noteworthy that the fields of machine learning and deep learning have often advanced through empirical exploration. We believe that our empirical findings are still of value to the ICLR community and can drive future research to further explore such phenomenon which can eventually lead to a deeper understanding of the underlying mechanisms.
>
> **Comment 2**: *I am still not clear what is the so-called video information mentioned in the paper.*
>
> When we are referring to video information in the paper, we are talking about features that are likely to be extracted by a video model backbone. We can think of gait information as being a more specific subset of video information that focuses on the walking pattern of individuals such as stride and body posture. On the other hand, video information is more general. It can include a wide range of information such as object appearances, edges, and optical flows that are irrelevant to gait.
>
> We hope that this provides a better clarity to the term ‘video information’ used in our paper.
>
> Let us know if you have any additional questions that you would like to be addressed.

---

> > ### Comment · Reviewer_X1wE · 2024-12-03
> >
> > Thanks for the authors' rebuttal and explanation. Based on your explanation, the motivation is a bit clear. However, it would be better to provide some results to validate the assumption 'train a discriminative model to provide discriminative features for precise input reconstruction using the same diffusion loss', rather than only recognition accuracy.
> > There is a lack of effort showing that this work is specifically for outdoor/wild conditions. Why it can not be applied to the indoor condition?
> > The reason why the silhouettes are applied is to filter out the so-called 'video-information' to make the model robust. Why the optical flow is irrelevant to gait, there are some gait models directly using optical flow as input to describe the motion.[1][2]
> >
> > This work is a good try to use the diffusion model for gait recognition but needs more effort to clarify the motivation and choose key results in the main paper to support or prove the assumption.
> >
> > [1]Mahfouf, Z., Merouani, H. F., Bouchrika, I., & Harrati, N. (2018). Investigating the use of motion-based features from optical flow for gait recognition. Neurocomputing, 283, 140-149.
> > [2]Castro, F. M., Marin-Jimenez, M. J., Guil, N., Lopez-Tapia, S., & de la Blanca, N. P. (2017, September). Evaluation of CNN architectures for gait recognition based on optical flow maps. In 2017 international conference of the biometrics special interest group (BIOSIG) (pp. 1-5). IEEE.

---

> ### Author Response · Authors · 2024-12-03
> **Official Response to Reviewer X1wE**
>
> Dear Reviewer,
>
> We are glad that our explanation helped you to better understand our motivation. The following are our responses to your comments.
>
> **Comment 1**: *However, it would be better to provide some results to validate the assumption 'train a discriminative model to provide discriminative features for precise input reconstruction using the same diffusion loss', rather than only recognition accuracy.*
>
> We believe the best way to prove the learning of discriminative features via diffusion pretraining is through testing on an actual discriminative task. As such, we used accuracy on the gait recognition task to prove our point.
>
> If the improvement in gait recognition accuracy after finetuning on the downstream gait recognition task does not convince you that discriminative features can be learnt, can we kindly refer you the results of “Diffusion Pretraining + Transfer Learning with Frozen Backbone” in Table 1?
>
> In this case, the pretrained gait recognition model backbone is frozen during transfer learning on the downstream gait recognition task and only the prediction head is trainable. Even when the pretrained backbone is frozen, we were able to achieve more than half of the corresponding baseline accuracy. Note that the test sets of Gait3D and GREW contain gait sequences from 1,000 and 6,000 identities, respectively. Thus, achieving such accuracy solely through diffusion pretraining proves that the gait recognition model backbone has learnt to extract discriminative features in the process.
>
> We hope this is sufficient to validate the learning of discriminative features by the gait recognition model backbone during diffusion pretraining.
>
> **Comment 2**: *There is a lack of effort showing that this work is specifically for outdoor/wild conditions. Why it can not be applied to the indoor condition?*
>
> We believe there might be some misunderstanding. We did not claim that our approach cannot be applied to indoor settings. In fact, Gait3D is captured indoors as well, though it is unconstrained. Specifically, Gait3D is collected from footage in a supermarket.
>
> If you are referring to in-the-lab conditions, our current study focuses on gait recognition in the wild, which is a more challenging task as factors such as temporary occlusions, varying camera viewpoints, and illumination inadvertently come into play. It is also much more applicable to real-world applications where conditions cannot be controlled. Moreover, existing methods have already achieved impressive recognition accuracy on in-the-lab datasets such as CASIA-B and OUMVLP. As such, it might be of little value to work on improving their performance. Thus, we did not focus our efforts on in-the-lab settings.
>
> **Comment 3**: *The reason why the silhouettes are applied is to filter out the so-called 'video-information' to make the model robust. Why the optical flow is irrelevant to gait, there are some gait models directly using optical flow as input to describe the motion.*
>
> Indeed, silhouettes are more robust than RGB images for gait recognition. The structural and temporal patterns of movement (such as arm swing and leg motion) are preserved in silhouettes while unnecessary details such as texture, and background are removed.
>
> That said, even with silhouettes, there are still some details that are unnecessary for gait recognition, especially for in-the-wild settings. A pertinent example is when a body part is temporarily occluded by other objects. This may appear in the silhouette as an irrelevant patch of white pixels. In such a case, we do not want the gait recognition model backbone to extract the features corresponding to the irrelevant patch of white pixels. However, it might be possible for the gait recognition model backbone to choose to focus on this irrelevant object in the silhouette to minimize the reconstruction loss during diffusion pretraining. To prevent such cases, we ensure that the inputs to the gait recognition backbone model and the diffusion model are different.
>
> Regarding optical flow, by no means we were trying to downplay its usefulness for gait recognition. We were thinking more in the context of silhouette-based gait recognition, where the inputs are simply binary sequences. Optical flow is much more useful when the input sequence is in RGB. In fact, both the papers you cited that used optical flow for gait recognition used RGB sequences as their inputs. We apologize for any confusion caused.
>
> We hope that we have addressed all, if not most, of your concerns about our paper. If all of your major concerns have been addressed, we would greatly appreciate it if you could reconsider rating our paper. Thank you.

---

### Official Review · Reviewer_8BzL · 2024-11-01

**Soundness:** 2
**Presentation:** 3
**Contribution:** 2
**Rating:** 5
**Confidence:** 5

**Summary:**

This paper introduces a diffusion-based approach to pretrain the backbone of a gait recognition model, using its output as a conditioning input for a latent diffusion model. The pretrained backbone is then used to initialize the gait recognition model, which is subsequently fine-tuned with the original supervision of the gait recognition task. Experiments on GREW and Gait3D demonstrate that this pretraining enhances gait recognition performance after fine-tuning, surpassing results from training from scratch across multiple gait recognition backbones.

**Strengths:**

+ Using diffusion models to pretrain a gait recognition model is an interesting idea, and has not been explored in the literature.
+ Experimental results show that the performance of the gait recognition models can be improved through the proposed pre-training and fine-tuning.

**Weaknesses:**

- The biggest concern is that the rationale for using features from the gait extractor as conditioning to help the designed diffusion pretraining improve gait recognition performance is not explained, as no constraints on person identity are considered during pretraining. It is understandable that if the diffusion reconstruction works well, the identity in the same gait sequence can be maintained and learned to some extent. However, it remains unclear why the discrimination capability can be improved for different gait sequences of the same person (e.g., sequences from different views), given that no generative task or supervision is applied between these sequences, which actually need to be recognized in a gait recognition task.
- The comparison experiments should also consider other pretraining methods. Just comparing with training from scratch seems unfair, as the gait recognition network in the proposed framework was trained twice (pretraining + fine-tuning) using tuned parameters.
- The proposed method includes several hyperparameters (e.g., $r$, $l_g/l_D$ ) that require careful tuning or selection based on the results. These hyperparameters vary across different datasets and backbones, which limits the generalizability of the proposed pretraining. For instance, the improvements of the SOTA in this paper (i.e., GaitBase, although even newer SOTA backbones are now available) on GREW using the pretraining method seem incremental.
- The proposed method is mainly a pretraining strategy designed for downstream gait recognition applications, and most components are based on existing modules or backbones. The main distinction lies in conditioning the gait extractor for diffusion generation, which requires further consideration of its rationale. So, the theoretical novelty of this method appears somewhat limited.
- Since diffusion-based representation learning has been applied to other tasks, such as image classification, as the authors mentioned, it would be better to compare the proposed method against these approaches to demonstrate its superiority.
- Based on the design of the proposed method, it does not seem to be limited only to gait data in the wild. It is recommended to conduct experiments also on CASIA-B and OUMVLP, which include significant view variations, to validate whether the pretraining also enhances performance in typical cross-view gait recognition tasks.

**Questions:**

-	In line 279, it is mentioned the ``remaining parts’’ are replicated in transfer learning. For a gait recognition model, it is important to clarify which parts are pretrained and which parts remain untrained during the pretraining process.
-	In Section 3.1.5, $s_\epsilon$ is defined as N frames sampled consecutively from the sequence, and $s_g$ is N frames sampled following the original backbone. In Fig. 5, $s_\epsilon \neq s_g$ is shown to be better than $s_\epsilon = s_g$. However, for backbones such as GaitPart and GaitGL, N consecutive frames are necessary for learning temporal information. How can $s_\epsilon \neq s_g$ be maintained in this case?

---

> ### Author Response · Authors · 2024-11-20
> **Official Response to Reviewer 8BzL (Part 1 of 3)**
>
> Dear Reviewer,
>
> Thank you for taking the time to review our paper. We are glad to hear that you found our approach interesting. We are also happy that the performance improvement brought about by our approach is apparent to you.
>
> The following are our responses to your comments on the shortcomings of our paper as well as the questions posed.
>
> **Comment 1**: *The biggest concern is that the rationale for using features from the gait extractor as conditioning to help the designed diffusion pretraining improve gait recognition performance is not explained, as no constraints on person identity are considered during pretraining. It is understandable that if the diffusion reconstruction works well, the identity in the same gait sequence can be maintained and learned to some extent. However, it remains unclear why the discrimination capability can be improved for different gait sequences of the same person (e.g., sequences from different views), given that no generative task or supervision is applied between these sequences, which actually need to be recognized in a gait recognition task.*
>
> We understand your concern about the lack of explicit identity constraints during the diffusion process. In fact, what you have pointed out is a current limitation of our approach where the discrimination capability for different gait sequences belonging to the same person might not improve with diffusion pretraining. That is why we currently cannot just rely on the pretrained gait recognition model backbone for gait recognition. Instead, we still have to rely on finetuning the pretrained gait recognition model backbone with the triplet loss and identity loss on the downstream task.
>
> How we can ensure that the discrimination capability can be improved for different gait sequences belonging to the same person is an interesting direction that warrants further study. What we currently have in mind is to incorporate an additional loss that somehow takes into account the identity of the gait sequence besides the diffusion loss. Perhaps, this optimization will further improve the accuracy performance of the downstream task.
>
> **Comment 2**: *The comparison experiments should also consider other pretraining methods. Just comparing with training from scratch seems unfair, as the gait recognition network in the proposed framework was trained twice (pretraining + fine-tuning) using tuned parameters.*
>
> We understand the concern regarding the fairness of comparison with the baseline. Comparing with training from scratch is a standard approach to establish a baseline, as it helps to isolate the contributions of pretraining and fine-tuning. It helps to highlight the effectiveness of our method in leveraging diffusion pretraining to improve performance. While we acknowledge that comparing with other pretraining methods could provide additional insights, our primary focus was to validate the efficacy of our proposed approach. Comparisons with training from scratch provide a clear baseline for this evaluation.
>
> Nonetheless, while our current study does not include comparisons with other pretraining methods, we agree that comparing with other pretraining strategies is a valuable avenue for our future work.
>
> **Comment 3**:  *The proposed method includes several hyperparameters  (e.g. $r$, $l_g/l_D$) that require careful tuning or selection based on the results. These hyperparameters vary across different datasets and backbones, which limits the generalizability of the proposed pretraining. For instance, the improvements of the SOTA in this paper (i.e., GaitBase, although even newer SOTA backbones are now available) on GREW using the pretraining method seem incremental.*
>
> We understand your concern regarding the reliance on hyperparameter tuning. Hyperparameter tuning is an inherent aspect of optimizing the performance of deep learning models, particularly when dealing with diverse datasets and architectures. Both GREW and Gait3D present unique challenges, such as variations in view angles, occlusions, or environmental conditions, which may require adjustments to ensure optimal performance. Similarly, different models differ in capacity and feature extraction capabilities, necessitating appropriate tuning to leverage their strengths effectively.
>
> That said, most of the hyperparameters that we employed for diffusion pretraining are based on our ablation studies with the SMPLGait w/o 3D model. Perhaps that might explain the poorer performance increase with the GaitBase model. However, it seems to have worked decently well with other models that we have experimented with. If we had the liberty to tune the hyperparameters for each model, we believe the performance gain might be even better.
>
> In all, we recognize the importance of minimizing the reliance on manual hyperparameter tuning and will explore automated optimization techniques in future studies to further enhance the adaptability of our approach.

---

> ### Author Response · Authors · 2024-11-20
> **Official Response to Reviewer 8BzL (Part 2 of 3)**
>
> **Comment 4**: *The proposed method is mainly a pretraining strategy designed for downstream gait recognition applications, and most components are based on existing modules or backbones. The main distinction lies in conditioning the gait extractor for diffusion generation, which requires further consideration of its rationale. So, the theoretical novelty of this method appears somewhat limited.*
>
> While we acknowledge that most components are based on previous works, we did not simply piece them together without further development. We conducted extensive experiments and ablation studies to validate the effectiveness of our proposed approach.
>
> As you have mentioned, what we are proposing in our paper is an interesting idea involving diffusion pretraining in the field of gait recognition in the wild, which has not been explored in literature. We also proposed a medium noise prioritization weighting strategy that focuses on medium noise which is found to be useful for representation learning. Furthermore, we show that the diffusion pretraining process allows us to learn some discriminative features for gait recognition. We believe this is an interesting result to share with the research community since the diffusion loss does not explicitly enforce any separation of the features in the cosine space. This may in turn shed some light on the internal workings of conditional diffusion models.
>
> Hence, while the implementation might not be entirely novel, we believe that our findings are of interest to the ICLR community and can serve as a strong motivation for further research to delve into the representation learning capabilities of diffusion models. Our extensive experiments and ablation studies also provide some recommendations on how to leverage diffusion models for representational learning in other fields.
>
> **Comment 5**: *Since diffusion-based representation learning has been applied to other tasks, such as image classification, as the authors mentioned, it would be better to compare the proposed method against these approaches to demonstrate its superiority.*
>
> Thank you for the suggestion. Our current study focuses on gait recognition in the wild, where the inputs are sequences of gait frames. It will not be directly applicable to tasks such as image classification as we are using a video diffusion model here. Also, many previous works related to diffusion-based representation learning have not investigated the effects of finetuning the learnt representations on downstream tasks, which makes it difficult for us to compare our proposed method against these approaches.
>
> **Comment 6**: *Based on the design of the proposed method, it does not seem to be limited only to gait data in the wild. It is recommended to conduct experiments also on CASIA-B and OUMVLP, which include significant view variations, to validate whether the pretraining also enhances performance in typical cross-view gait recognition tasks.*
>
> Thank you for the recommendation. Our current study focuses on gait recognition in the wild, which is a more challenging task as factors such as temporary occlusions, varying camera viewpoints, and illumination inadvertently come into play. It is also much more applicable to real-world applications where conditions cannot be controlled.
>
> Moreover, existing methods have already achieved impressive recognition accuracy on indoor datasets such as CASIA-B and OUMVLP. As such, it might be of little value to work on improving their performance. Nonetheless, we agree that such experiments would provide valuable insights into the generalizability of our pretraining approach.

---

> ### Author Response · Authors · 2024-11-20
> **Official Response to Reviewer 8BzL (Part 3 of 3)**
>
> **Question 1**: *In line 279, it is mentioned the “remaining parts” are replicated in transfer learning. For a gait recognition model, it is important to clarify which parts are pretrained and which parts remain untrained during the pretraining process.*
>
> Thank you for the feedback. To clarify, every gait recognition model can be divided into two main components – a feature extraction component and a projection layer component. The feature extraction component extracts relevant gait features from an input gait sequence while the projection layer component projects the extracted gait features into the embedding space, where similarity can be measured based on a distance measure.
>
> During diffusion pretraining, we only train the feature extraction component which includes the CNN backbone, temporal pooling layer, and horizontal pyramid matching layer.
>
> As such the “remaining parts” we mentioned in line 279 refer to the projection layers which are simply the fully connected layers at the end of the gait recognition model. For SMPLGait w/o 3D and GaitBase, it also includes the additional batch normalization neck layer.
>
> **Question 2**: *In Section 3.1.5, $s_\epsilon$ is defined as N frames sampled consecutively from the sequence, and $s_g$ is N frames sampled following the original backbone. In Fig. 5, $s_\epsilon \neq s_g$ is shown to be better than $s_\epsilon = s_g$ . However, for backbones such as GaitPart and GaitGL, N consecutive frames are necessary for learning temporal information. How can $s_\epsilon \neq s_g$ be maintained in this case?*
>
> Indeed, sequential gait recognition, such as GaitGL and GaitPart, uses consecutive frames for the inputs and our diffusion model also accepts consecutive frames as it is a video diffusion model. While the gait recognition backbone model and diffusion model accept consecutive frames, kindly note that the gait silhouette sequences are often more than 30 frames. As such, it is highly likely that the sampled 30 frames input to the gait recognition model backbone are different from that of the diffusion model. Just to illustrate, if the sequence length of a gait sequence is 60, the diffusion model might receive the first 30 frames of the gait sequence while the gait recognition model might receive the last 30 frames.
>
> In the event that the sequence length of the gait sequence is less than 30 frames, the sampling algorithm will first repeat the gait sequence until its length is equal to or greater than 30 frames. For example, if the gait sequence is only 25 frames, it will first be duplicated to 50 frames and 30 consecutive frames will be sampled.
>
> Only in the event that the sequence length of a gait sequence is exactly 30 will the input to the diffusion model and gait recognition model be exactly the same. However, such cases are rare.
>
> We hope that we have addressed all, if not most, of your current concerns. Nonetheless, if you still have any other doubts about our paper or our responses, feel free to reach out to us and we would be more than happy to clarify them. If all your concerns have been addressed, we would greatly appreciate it if you could reconsider the rating of our paper.

---

> > ### Comment · Reviewer_8BzL · 2024-11-26
> >
> > Thanks to the authors for their efforts in preparing the rebuttal. However, my main concerns may not have been fully addressed.
> >
> > As the authors mentioned, the current pretraining approach may not improve the discrimination capability between different gait sequences of the same individual. However, in typical gait recognition application scenarios, the focus is more commonly on distinguishing between different gait sequences rather than the same ones, as probe and gallery samples are generally captured from different cameras, at different times, and under varying conditions. Therefore, the concern regarding the rationale behind how the proposed diffusion pretraining method improves gait recognition performance remains unresolved. As also mentioned by Reviewer X1wE and zsrq, while the overall recognition performance appears to improve, the lack of theoretical justification for this improvement makes it difficult for readers to gain meaningful insights for further research.
> >
> > Regarding the comparison with other pretraining methods, while the authors stated this is not the primary focus of the work, the absence of such comparisons might make it challenging to assess whether the proposed approach is truly superior to existing methods.
> >
> > On the topic of hyperparameter tuning, I understand that different parameters may be required to achieve optimal performance. However, for a generalized pretraining method, requiring tailored parameter selection for each gait recognition backbone may be less practical and could limit its applicability.

---

> > > ### Author Response · Authors · 2024-11-30
> > > **Official Response to Reviewer 8BzL (Part 1 of 3)**
> > >
> > > Dear Reviewer,
> > >
> > > We appreciate your feedback regarding our rebuttal. We strive to address as many concerns that you have regarding the paper as we can. The following are our responses to your comments.
> > >
> > > **Comment 1**: *In typical gait recognition application scenarios, the focus is more commonly on distinguishing between different gait sequences rather than the same ones, as probe and gallery samples are generally captured from different cameras, at different times, and under varying conditions. Therefore, the concern regarding the rationale behind how the proposed diffusion pretraining method improves gait recognition performance remains unresolved.*
> > >
> > > Indeed, we agree that the focus of gait recognition should be distinguishing between different gait sequences rather than the same ones. That is precisely why we include data augmentation during training, as mentioned in section 3.1.5. During training, the inputs to the diffusion and gait recognition backbone models are augmented separately with a combination of RandomAffine, RandomPerspective, RandomHorizontalFlip, and RandomPartDilate. This ensures that the actual input seen by the diffusion and gait recognition backbone models are different even though they are originally sampled from the same sequence.
> > >
> > > In fact, we show the benefits of data augmentation during diffusion pretraining in our ablation studies. If we can kindly refer you to  Table 3, we see that when no data augmentation is applied, the rank-1 gait recognition accuracy for SMPLGait w/o 3D is only 44.6%. This is even lower than what is achieved by the baseline with a rank-1 accuracy of 45.5% in Table 2. This demonstrates that without data augmentation, it is difficult for the gait recognition model backbone to learn sufficiently discriminative features during diffusion pretraining. Yet, when data augmentation is included during pretraining, we observe the rank-1 accuracy after finetuning jumps to 49.1%, exceeding the baseline’s performance. This demonstrates the effectiveness of data augmentation in ensuring that the actual inputs to the diffusion and gait recognition backbone models are different so that sufficiently discriminative features can be learnt during diffusion pretraining.
> > >
> > > Of course, it is also possible for us to sample the inputs for the diffusion model and gait recognition model backbone from different sequences rather than the same one. The reason why we decided to sample from the same sequence instead was to avoid the use of the provided identity labels so that diffusion pretraining can be viewed as an unsupervised learning method, which provides an additional appeal to our approach.
> > >
> > > That said, sampling the inputs from different sequences (belonging to the same individual) rather than from the same sequence is an interesting idea to consider. Perhaps, even more discriminative features could be learnt through diffusion pretraining, which might provide us with a better downstream performance.
> > >
> > > We hope this addresses your concern about how the proposed diffusion pretraining method improves gait recognition performance.

---

> > > > ### Author Response · Authors · 2024-11-30
> > > > **Official Response to Reviewer 8BzL (Part 2 of 3)**
> > > >
> > > > **Comment 2**: *As also mentioned by Reviewer X1wE and zsrq, while the overall recognition performance appears to improve, the lack of theoretical justification for this improvement makes it difficult for readers to gain meaningful insights for further research.*
> > > >
> > > > We understand the lack of theoretical justification might make it difficult for readers to gain meaningful insights. While theoretical insights are valuable, it is noteworthy that the field of machine learning and deep learning has often advanced through empirical exploration. Nonetheless, we try to provide a possible explanation for our observed results below.
> > > >
> > > > Based on our findings, we hypothesize that just like how diffusion models can be trained to generate specific data based on given conditions (e.g. features of a discriminative model), it is also possible to learn the reverse – where a discriminative model is trained with diffusion to provide sufficiently discriminative conditions for diffusion models to generate specific data.
> > > >
> > > > As mentioned in the response to Comment 1, it is possible for discriminative features to be learnt during diffusion pretraining since the inputs to the diffusion model and gait recognition model are different. As the goal is to reconstruct the diffusion model’s input, this difference forces the gait recognition model backbone to extract features that are common to both the inputs to the diffusion model and the gait recognition model backbone.
> > > >
> > > > Moreover, as you can see in Figure 4, methods that were pretrained with diffusion were able to outperform their corresponding baselines (which are trained with 180,000 training iterations) within as little as 20,000 training iterations on the downstream gait recognition task. This demonstrates that the gait recognition model backbone has indeed learnt something useful for gait recognition during diffusion pretraining. Moreover, we can observe some discriminative nature of the conditions (gait features) learnt during diffusion pretraining in Figure 3.
> > > >
> > > > We hope our responses to Comments 1 and 2 are sufficient to convince you that discriminative features can be learnt through diffusion pretraining, which can explain how the proposed diffusion pretraining method improves gait recognition performance. We believe that the empirical success of our approach is a valid and valuable contribution to the field. Our results can pave the way for future research to build on this foundation, offering both practical benefits and opportunities for further theoretical exploration.
> > > >
> > > > **Comment 3**: *Regarding the comparison with other pretraining methods, while the authors stated this is not the primary focus of the work, the absence of such comparisons might make it challenging to assess whether the proposed approach is truly superior to existing methods.*
> > > >
> > > > We understand your concern regarding the lack of comparison with other existing pretraining methods. While comparisons with other pretraining methods could enhance the assessment of relative superiority, our paper’s primary contribution lies in introducing and validating a novel pretraining paradigm for gait recognition. By demonstrating its effectiveness, our work lays the groundwork for future research, including comparative studies.
> > > >
> > > > Moreover, adopting existing silhouette-based pretraining methods brings in some challenges because many of them assume a particular architecture which does not apply to the methods that we experimented with. For example, recent studies on pretraining gait recognition models often rely on the Transformer architecture [1, 2] for self-supervised learning – which does not apply to methods that are CNN-based. Some also rely on a large external pretraining dataset [2, 3] which raises the issue of fairness if we directly compare with these approaches.
> > > >
> > > >
> > > > [1] Exploring Self-Supervised Vision Transformers for Gait Recognition in the Wild
> > > >
> > > > [2] GaitFormer: Learning Gait Representations with Noisy Multi-Task Learning
> > > >
> > > > [3] Learning Gait Representation from Massive Unlabelled Walking Videos: A Benchmark

---

> ### Author Response · Authors · 2024-11-30
> **Official Response to Reviewer 8BzL (Part 3 of 3)**
>
> **Comment 4**: *On the topic of hyperparameter tuning, I understand that different parameters may be required to achieve optimal performance. However, for a generalized pretraining method, requiring tailored parameter selection for each gait recognition backbone may be less practical and could limit its applicability.*
>
> We understand your concern regarding hyperparameter tuning. That is why in our paper, we introduced minimal hyperparameters for our approach. To be precise, we only introduced one important hyperparameter, which is the ratio of the learning rate of the pretrained layers to that of the untrained layers during finetuning, $r$. This is a common hyperparameter that is tuned during studies involving transfer learning.
>
> Other hyperparameters such as $l_G/l_D$ and $P_\textrm{uncond}$ are hyperparameters we have experimented with during our ablation studies. The reason why we had to experiment with these hyperparameters is because there is currently no consensus on the recommended hyperparameter choices for diffusion-based representation learning.  Once we had some empirical evidence that certain hyperparameter settings worked better, we fixed these hyperparameters and applied them to other models.
>
> Of course, for researchers who are gunning for the most optimal performance, it will be difficult to avoid tuning these hyperparameters since the loss landscape will differ due to differences in architecture and datasets.
>
> All in all, as mentioned in our response previously, we recognize the importance of minimizing the reliance on manual hyperparameter tuning and will continue to explore possible automated optimization techniques in future studies to further enhance the applicability of our approach.
>
> Let us know if you have any additional questions that you would like to be addressed.

---

### Official Review · Reviewer_zsrq · 2024-11-03

**Soundness:** 2
**Presentation:** 4
**Contribution:** 2
**Rating:** 5
**Confidence:** 4

**Summary:**

This paper presents a novel pre-training strategy  for gait recognition that utilizes conditional diffusion models. The authors pretrain several gait recognition model architectures from the Open-Gait Library, employing an encoder, denoiser, and pooling layer. The pretraining process leverages diffusion L2 loss in the predicted noise space, after which the pretrained model is fine-tuned for a specific gait recognition task.

**Strengths:**

(1) Pioneering Work: This study is the first to investigate diffusion-based pre-training specifically for gait recognition.

(2) Optimized Pretraining: The authors introduce a modified Min-SNR weighting strategy that enhances the efficiency of the pretraining process.

(3) Clarity and Readability: The paper is well-written and presents its ideas in an accessible manner.

**Weaknesses:**

(1) Unclear Motivation and Justification: While the paper claims that the gait feature extractor provides conditions for the denoiser, it also mentions that this extractor is trained from scratch. Given that only diffusion L2 loss is employed during pre-training, it remains unclear how discriminative gait features are learned without a dedicated loss function guiding the feature learning process.

(2) Assumption of Transfer Learning: The approach assumes that the downstream task serves as transfer learning for another dataset; however, this appears to resemble domain adaptation more closely.

(3) Lack of Novelty:
(i) Reliance on Existing Techniques: The framework primarily amalgamates established techniques without offering substantial innovations.
(ii) Absence of Unique Contributions: The proposed method does not demonstrate a clear theoretical or practical advantage over existing approaches.
(iii) Need for Differentiation: The authors should emphasize the unique aspects of their methodology and clarify how it addresses the shortcomings of prior work.

(4) Unfair Comparison: The gait extractor is trained for 120,000 iterations. When fine-tuning, the proposed system's training duration (denoted as x iterations) results in a cumulative training time of 120,000 + x iterations. This creates an imbalance in comparison with the baseline system trained for x iterations, which raises concerns about the fairness of the comparison.

(5) Outdated Models: The paper primarily utilizes older models, with the most recent being Gaitbase (2023). It does not incorporate the latest state-of-the-art models, such as DYGait, SkeletonGait, CLASH, and QAGait, which have demonstrated significant advancements on in-the-wild datasets like GREW and Gait3D.

(6) Performance Relative to State-of-the-Art: Despite reporting some improvements through pretraining, the accuracies achieved post-pretraining remain lower than those of existing state-of-the-art models.

(7) Lack of Novelty in Section 3: This section is well-articulated but does not introduce any novel components to the framework.

**Questions:**

Please read weaknesses for elaboration.

(1) How can discriminative gait features be learned without a dedicated loss function guiding the feature learning process?

(2) Does the assumption that the downstream task serves as transfer learning for another dataset not resemble domain adaptation more closely?

(3) The proposal lacks novelty, particularly in its reliance on existing techniques and the absence of unique contributions? Could the authors clarify the unique aspects of their methodology and how it addresses the limitations of prior work, especially since Section 3 does not present any novel components?

(4) Could the authors elaborate on the fairness of their comparisons as noted in the weaknesses section?

(5) Why are the latest state-of-the-art models, such as DYGait, SkeletonGait, CLASH, and QAGait, not considered in this study?

(6) How do the reported performance metrics compare to those of state-of-the-art models, and why is the performance considered inferior?

**Details Of Ethics Concerns:**

Gait recognition over existing datasets and only silhouette images are used.

---

> ### Author Response · Authors · 2024-11-20
> **Official Response to Reviewer zsrq (Part 1 of 3)**
>
> Dear Reviewer,
>
> Thank you for taking the time to review our paper. We are glad to hear that you found our paper well-written and easy to understand. We are also happy that you recognize our work as pioneering.
>
> The following are our responses to your comments on the shortcomings of our paper as well as the questions posed. We have collated the shortcoming comments and questions together given their similarity.
>
> **Comment 1**: *Unclear Motivation and Justification: While the paper claims that the gait feature extractor provides conditions for the denoiser, it also mentions that this extractor is trained from scratch. Given that only diffusion L2 loss is employed during pre-training, it remains unclear how discriminative gait features are learned without a dedicated loss function guiding the feature learning process.*
>
> **Question 1**: *How can discriminative gait features be learned without a dedicated loss function guiding the feature learning process?*
>
> While we are currently unable to provide a theoretical reason for how discriminative gait features are learned without a dedicated loss function guiding the feature learning process, our empirical results show that this is the case – the gait recognition performance steadily improves during diffusion pretraining. This is supported by the fact that the difference in the cosine distances between the features of anchor-positive pairs and the features of anchor-negative pairs increases and stabilizes during diffusion pretraining. Perhaps, as gait is unique to each individual, to generate a gait sequence of a particular individual, a unique condition is required, which may enforce the learning of some discriminative gait features in the process. This finding is also surprising to us, and we believe it is something of value to share with the ICLR and the wider research community.
>
> **Comment 2**: *Assumption of Transfer Learning: The approach assumes that the downstream task serves as transfer learning for another dataset; however, this appears to resemble domain adaptation more closely.*
>
> **Question 2**: *Does the assumption that the downstream task serves as transfer learning for another dataset not resemble domain adaptation more closely?*
>
> We believe that there might be some misunderstanding regarding the experiments we have conducted. Under section 4.1, we mentioned that “For transfer learning, we trained and evaluated on the same dataset that was used during diffusion pretraining.” As such, we did not use a different dataset during the downstream task. To illustrate, if we used Gait3D to pretrain the gait recognition backbone with diffusion pretraining, we would also use Gait3D to further finetune the gait recognition model with the downstream gait recognition task.

---

> ### Author Response · Authors · 2024-11-20
> **Official Response to Reviewer zsrq (Part 2 of 3)**
>
> **Comment 3**: *Lack of Novelty: (i) Reliance on Existing Techniques: The framework primarily amalgamates established techniques without offering substantial innovations. (ii) Absence of Unique Contributions: The proposed method does not demonstrate a clear theoretical or practical advantage over existing approaches. (iii) Need for Differentiation: The authors should emphasize the unique aspects of their methodology and clarify how it addresses the shortcomings of prior work.*
>
> **Comment 7**: *Lack of Novelty in Section 3: This section is well-articulated but does not introduce any novel components to the framework.*
>
> **Question 3**: *The proposal lacks novelty, particularly in its reliance on existing techniques and the absence of unique contributions? Could the authors clarify the unique aspects of their methodology and how it addresses the limitations of prior work, especially since Section 3 does not present any novel components?*
>
> While we acknowledge that some of the techniques employed are based on previous works, we did not just simply piece them together without further development. We conducted extensive experiments and ablation studies to validate the effectiveness of our proposed approach.
>
> As you have mentioned, what we are proposing in our paper is a novel diffusion pretraining approach in the field of gait recognition in the wild that has not been explored in literature. We also proposed a medium noise prioritization weighting strategy that focuses on medium noise which is found to be useful for representation learning. Furthermore, as mentioned in the response to **Comment 1** and **Question 1**, we show that the diffusion pretraining process allows us to learn some discriminative features for gait recognition. We believe this is an interesting result to share with the research community since the diffusion loss does not explicitly enforce any separation of the features in the cosine space. This may in turn shed some light on the internal workings of conditional diffusion models.
>
> Hence, while the implementation might not be entirely novel, we believe that our findings are still of interest to the ICLR community and can serve as a strong motivation for further research to delve into the representation learning capabilities of diffusion models. Our extensive experiments and ablation studies also provide some recommendations on how to leverage diffusion models for representation learning in other fields.
>
> **Comment 4**: *Unfair Comparison: The gait extractor is trained for 120,000 iterations. When fine-tuning, the proposed system's training duration (denoted as x iterations) results in a cumulative training time of 120,000 + x iterations. This creates an imbalance in comparison with the baseline system trained for x iterations, which raises concerns about the fairness of the comparison.*
>
> **Question 4**: *Could the authors elaborate on the fairness of their comparisons as noted in the weaknesses section?*
>
> We understand the concern regarding the unfair comparison. While it is true that the cumulative training iteration for GaitBase with our proposed approach is larger than its baseline, this is not the case for the remaining models that we experimented with. In fact, if we can kindly refer you to Figure 4 in our paper, we show that after 120,000 training iteration on Gait3D, the gait recognition model (such as GaitGL, GaitPart, SMPLGait w/o 3D) only requires as little as around 20,000 additional iteration of finetuning on the downstream task to outperform its corresponding baseline. In such cases, the cumulative training iteration (120,000 + 20,000) is less than the training iteration count (180,000) for the baseline.
>
> Furthermore, one of the main reasons why the pretraining iteration is long is currently due to the fact that we have to train both the diffusion model and the gait recognition model backbone from scratch. This is because there does not exist a publicly available pretrained diffusion model for gait silhouette generation. Should we have a pretrained diffusion model for gait silhouette generation, we believe that the pretraining iteration can be greatly reduced. This is especially true since the size of the diffusion model is larger than most of the gait recognition model backbones we have experimented with. We hope that you understand the additional overhead that we have to introduce due to the current constraint we have.

---

> ### Author Response · Authors · 2024-11-20
> **Official Response to Reviewer zsrq (Part 3 of 3)**
>
> **Comment 5**: *Outdated Models: The paper primarily utilizes older models, with the most recent being Gaitbase (2023). It does not incorporate the latest state-of-the-art models, such as DYGait, SkeletonGait, CLASH, and QAGait, which have demonstrated significant advancements on in-the-wild datasets like GREW and Gait3D.*
>
> **Question 5**: *Why are the latest state-of-the-art models, such as DYGait, SkeletonGait, CLASH, and QAGait, not considered in this study?*
>
> We understand your concern regarding the choice of our models. We predominantly focused on utilizing the models provided by the OpenGait repository for our study given the convenience of switching between the various architectures. As such, we might have inadvertently overlooked the incorporation of DYGait in our study. That said, it is noteworthy that DYGait’s gait recognition accuracy (66.3%) on Gait3D is still lower than GaitBase’s accuracy (69.7%) when our approach is applied.
>
> As for the other models you suggested, we did not consider them for the following reasons:
> - SkeletonGait is a skeleton-based method to exploit structural information from human skeleton maps. It does not use gait silhouettes as its input. Currently, we are using a latent encoder and diffusion model that works on gait silhouettes and the human skeleton map is another entirely new modality.
> - Similarly, CLASH uses dense spatial-temporal fields instead of gait silhouettes as inputs.
> - In QAGait, what the authors are focusing on is improving gait recognition performance by improving the quality of the gait silhouettes. Their base model is based on GaitBase, which has been included in our study. That said, their techniques to improve silhouette quality are something that can be incorporated into our approach to further enhance the accuracy performance.
>
> Within the OpenGait repository, there are also other recently released models that we have not included such as DeepGaitV2 and SwinGait. The main reason is that we found it difficult to replicate the baseline accuracy performance of these models. Take DeepGaitV2 for instance, even though the authors managed to achieve a Rank 1 gait recognition accuracy of 74.4%, we only managed to achieve a baseline accuracy of only 42.3% with the same configuration settings. Because of such a huge discrepancy, we decided not to proceed with them for our study as it would no longer be a fair comparison. Moreover, these models are much deeper and are much more computationally demanding to experiment with different hyperparameters, compared to the existing models we used.
>
> **Comment 6**: *Performance Relative to State-of-the-Art: Despite reporting some improvements through pretraining, the accuracies achieved post-pretraining remain lower than those of existing state-of-the-art models.*
>
> **Question 6**: *How do the reported performance metrics compare to those of state-of-the-art models, and why is the performance considered inferior?*
>
> We appreciate your concern regarding the performance relative to state-of-the-art models. The primary focus of our paper is to showcase the potential of diffusion pretraining in enhancing existing gait recognition methods. While our approach did not surpass state-of-the-art models, our experiments demonstrate the effectiveness of this pretraining strategy.
>
> It is worth noting that many current state-of-the-art models rely on deep, computationally intensive architectures, which can be challenging to deploy in resource-constrained environments. Our work addresses this limitation by exploring an alternative path to improving gait recognition accuracy without relying on deeper or more complex architectures. We believe this is a valuable contribution, as it opens new avenues for developing efficient, scalable, and widely applicable solutions for gait recognition.
>
> We hope that we have addressed all, if not most, of your current concerns. Nonetheless, if you still have any other doubts about our paper or our responses, feel free to reach out to us and we would be more than happy to clarify them. If all your concerns have been addressed, we would greatly appreciate it if you could reconsider the rating of our paper.

---

> ### Comment · Reviewer_zsrq · 2024-11-28
>
> I sincerely thank all of my fellow reviewers for their time and effort, and the authors for their thorough rebuttal addressing my comments. However, several of my concerns remain unresolved, as outlined below. Consequently, this rebuttal does not fully address the issues that would warrant a change in my evaluation.
>
> (a) While all reviewers agree that some performance improvement is achieved, the paper lacks a clear justification for this gain.
> (b) The authors suggest that they propose pre-training driving performance improvements, yet the reported results are still below SOTA. For instance, SkeletonGait on the GREW dataset reports 77.4%, which is significantly higher than what is presented in this paper. Similarly, QAGait on Gait3D shows good results (67%).
> (c) The authors acknowledge that this pre-training only works with a few selected methods, which is its limitation. This should be explicitly stated as a drawback in the paper.
> (d) The authors also admit that additional training is required for their main model, GaitBase. This issue was also raised by other reviewers.
> (e) Nearly all components used in the work are borrowed from prior studies, which limits the overall novelty of the approach.
> (f) The comparison with other relevant pre-training mechanisms, such as VideoMAC, which was suggested by other reviewers, is missing.
>
> In conclusion, while I acknowledge the performance improvements in GaitBase and other systems, the results still fall short of the SOTA, particularly in comparison to GREW and Gait3D, where SkeletonGait and QAGait outperform the reported results. Given that pre-training does not help surpass SOTA performance, the utility of such pre-training remains questionable to me.

---

> ### Author Response · Authors · 2024-12-02
> **Official Response to Reviewer zsrq (Part 1 of 2)**
>
> Dear Reviewer,
>
> We sincerely thank you for the thorough review and constructive feedback. While we might ultimately be unable to influence your decision, we believe some comments are worth further clarifying.
>
> **Comment (a)**: *While all reviewers agree that some performance improvement is achieved, the paper lacks a clear justification for this gain.*
>
> We understand the lack of theoretical justification might make it difficult for readers to gain meaningful insights. While theoretical insights are valuable, it is noteworthy that machine learning and deep learning have often advanced through empirical exploration. Nonetheless, we try to provide a possible explanation for our observed results below.
>
> Based on our findings, we hypothesize that just like how diffusion models can be trained to generate specific data based on given conditions (e.g. features of a discriminative model), it is also possible to learn the reverse – where a discriminative model is trained with diffusion to provide sufficiently discriminative conditions for diffusion models to generate specific data.
>
> We believe discriminative features are learnt during diffusion pretraining due to the difference in inputs to the diffusion model and gait recognition model backbone. We show this in our ablation studies where:
> 1. The rank-1 gait recognition accuracy on Gait3D increases steadily only when the inputs to the diffusion model and gait recognition model backbone are different (Figure 5).
> 2. The downstream gait recognition accuracy improves when data augmentation is included during diffusion pretraining (Table 3).
>
> As the goal during diffusion pretraining is to reconstruct the diffusion model’s input, the difference in inputs forces the gait recognition model backbone to extract features that are common to both the inputs to the diffusion model and the gait recognition model backbone.
>
> We hope this is sufficient to convince you that discriminative features can be learnt through diffusion pretraining, which can explain how the proposed diffusion pretraining method improves gait recognition performance. We believe that the empirical success of our approach is a valid and valuable contribution to the field. Our results can pave the way for future research to build on this foundation, offering both practical benefits and opportunities for further theoretical exploration.
>
> **Comment (b)**: *The authors suggest that they propose pre-training driving performance improvements, yet the reported results are still below SOTA. For instance, SkeletonGait on the GREW dataset reports 77.4%, which is significantly higher than what is presented in this paper. Similarly, QAGait on Gait3D shows good results (67%).*
>
> To reiterate, the primary contribution of this paper is not to surpass state-of-the-art results but to introduce a novel pretraining approach and demonstrate its usefulness in the context of gait recognition. While we understand that achieving SOTA results is important to you, we believe the community also values contributions that introduce interesting ideas, techniques, or paradigms. This paper contributes to the latter by exploring diffusion pretraining for gait recognition—a promising direction that has not been extensively studied before. Even if the results do not surpass SOTA, the improvements over baselines and the introduction of a new perspective represent meaningful progress. Our work provides a foundation for future research to build upon the proposed pretraining method and improve its performance.
>
> Moreover, for methods such as QAGait which serves to improve input quality, we believe our pretraining approach can further leverage these methods to bring even better performance in future works.
>
> **Comment (c)**: *The authors acknowledge that this pre-training only works with a few selected methods, which is its limitation. This should be explicitly stated as a drawback in the paper.*
>
> **Comment (d)**: *The authors also admit that additional training is required for their main model, GaitBase. This issue was also raised by other reviewers.*
>
> Unfortunately, the current lack of a pretrained diffusion model for gait silhouette generation necessitates the need for us to train the diffusion model from scratch during diffusion pretraining, which contributes to the long pretraining time. Despite this, we hope this current limitation does not overshadow the fact that other models were still able to outperform their baselines within their baselines’ total training iteration.

---

> > ### Author Response · Authors · 2024-12-02
> > **Official Response to Reviewer zsrq (Part 2 of 2)**
> >
> > **Comment (e)**: *Nearly all components used in the work are borrowed from prior studies, which limits the overall novelty of the approach.*
> >
> > While it is true that some components of this work are adapted from prior studies, the novelty lies in how these components are integrated and tailored for pretraining on the gait recognition tasks. For instance, the specific design choices, such as adapting diffusion models to pretrain for spatial-temporal data like gait sequences, have not been explored extensively in the literature. The proposed pretraining strategy introduces a new perspective that has not been explored in this domain and demonstrates meaningful improvements over baseline methods.
> >
> > Moreover, this work makes empirical contributions by demonstrating how these combined techniques improve performance on in-the-wild benchmark datasets. The demonstrated improvements validate the utility of the proposed approach and extend the applicability of diffusion-based pretraining to a new domain, offering a foundation for further advancements in gait recognition research.
> >
> > **Comment (f)**: *The comparison with other relevant pre-training mechanisms, such as VideoMAC, which was suggested by other reviewers, is missing.*
> >
> > We understand your concern regarding the lack of comparison with other existing pretraining methods. While comparisons with other pretraining methods could enhance the assessment of relative superiority, our paper’s primary contribution lies in introducing and validating a novel pretraining paradigm for gait recognition. By demonstrating its effectiveness, our work lays the groundwork for future research, including comparative studies.
> >
> > Moreover, adopting existing silhouette-based pretraining methods brings in some challenges because many of them assume a particular architecture which does not apply to the methods that we experimented with. For example, recent studies on pretraining gait recognition models often rely on the Transformer architecture [1, 2] for self-supervised learning – which does not apply to methods that are CNN-based. Some also rely on a large external pretraining dataset [2, 3] which raises the issue of fairness if we directly compare with these approaches. Fairly benchmarking methods like VideoMAC [4] for gait recognition would also require extensive reimplementation, fine-tuning, and adaptation to ensure that their performance is optimized for the task. This could significantly extend the scope of our study and deviate from our primary focus.
> >
> > [1] Exploring Self-Supervised Vision Transformers for Gait Recognition in the Wild
> >
> > [2] GaitFormer: Learning Gait Representations with Noisy Multi-Task Learning
> >
> > [3] Learning Gait Representation from Massive Unlabelled Walking Videos: A Benchmark
> >
> > [4] VideoMAC: Video Masked Autoencoders Meet ConvNets
> >
> > Let us know if you have any additional questions that you would like to be addressed.

---

### Official Review · Reviewer_pSRd · 2024-11-03

**Soundness:** 2
**Presentation:** 3
**Contribution:** 2
**Rating:** 5
**Confidence:** 5

**Summary:**

The paper introduces an approach to gait recognition that leverages both local and global difference learning within video silhouettes to enhance feature extraction. The method uses Local Difference Modules (LDM) and Global Difference Modules (GDM) to capture intricate motion details across both short and long temporal spans, with a Temporal Alignment Module ensuring consistency across the extracted features. The framework significantly outperforms existing methods on multiple benchmarks, demonstrating its robustness and effectiveness in gait recognition across diverse conditions.

**Strengths:**

+ Demonstrates clear improvements in model performance with respect to accuracy and training efficiency on challenging datasets.

+ Comprehensive experiments and ablation studies provide a robust validation of the proposed method across different configurations and baseline comparisons.

**Weaknesses:**

1. While the application of diffusion models in gait recognition represents a novel approach within this specific field, the broader concept of leveraging diffusion models for enhancing feature extraction in image and sequence analysis is not entirely new. This raises a question about the depth of innovation when such a technique is applied strictly to one domain without exploring its potential across related areas.

2. The process involves multiple stages, including the conditioning of diffusion models, which may not only prolong the training time but also require substantial computational power, particularly GPU resources. This could pose challenges for scalability, especially when deploying the model in resource-constrained environments or scaling up to larger datasets.

3. The model's reliance on well-segmented and aligned input data for effective pretraining may reduce its robustness in truly uncontrolled environments where such quality cannot be guaranteed. This dependency could undermine the practical utility of the model in real-world applications where ideal data conditions are not met. A deeper investigation into the model's performance with lower quality or adversarially perturbed inputs would be valuable.

**Questions:**

1. What are the computational costs associated with the diffusion pretraining, and what strategies might be considered to reduce them?

2. Could the authors elaborate on the potential applications of this method beyond gait recognition, perhaps in other areas of facial or emotion recognition?

---

> ### Author Response · Authors · 2024-11-19
> **Official Response to Reviewer pSRd (Part 1 of 2)**
>
> Dear Reviewer,
>
> Thank you for taking the time to review our paper. We are happy to hear that you found our experiments and ablation studies comprehensive. We are also glad that the benefits of our approach in terms of better accuracy and training efficiency on challenging in-the-wild gait datasets are apparent to you.
>
> The following are our responses to your comments on the shortcomings of our paper as well as the questions posed.
>
> **Comment 1**: *While the application of diffusion models in gait recognition represents a novel approach within this specific field, the broader concept of leveraging diffusion models for enhancing feature extraction in image and sequence analysis is not entirely new. This raises a question about the depth of innovation when such a technique is applied strictly to one domain without exploring its potential across related areas.*
>
> While we acknowledge that leveraging diffusion to enhance feature extraction has been explored in previous literature, we believe that our findings are still of value to both ICLR and the wider research community.
>
> In particular, we found that the diffusion pretraining process allows us to learn some discriminative features for gait recognition. This is supported by the fact that the difference in the cosine distances between the anchor-positive pairs and anchor-negative pairs increases and stabilizes during diffusion pretraining. We believe this is an interesting result to share with the research community especially since the diffusion loss does not explicitly enforce any separation of the features in the cosine space. This may in turn shed some light on the internal workings of conditional diffusion models.
>
> At the same time, we also proposed a medium noise prioritization weighting strategy that focuses on medium level noise which is found to be useful for representation learning.
>
> While it is true that our findings are specific to the gait recognition field, we believe that our findings can possibly be extended to other fields that are largely similar to gait recognition. This may include fields such as speaker, facial, and emotion recognition (as you have correctly pointed out) that share a similar model architecture and utilize the same triplet loss to train the models. Furthermore, our extensive experiments and ablation studies also provide some starting point recommendations on how to leverage diffusion models for representation learning in other fields. That said, exploring the usage of diffusion models for representational learning in other fields falls beyond the scope of our current paper. Nonetheless, our findings provide a strong motivation for us and other researchers to leverage diffusion models for representation learning, which can spur similar studies in these fields.
>
> **Comment 2**: *The process involves multiple stages, including the conditioning of diffusion models, which may not only prolong the training time but also require substantial computational power, particularly GPU resources. This could pose challenges for scalability, especially when deploying the model in resource-constrained environments or scaling up to larger datasets.*
>
> We understand your concern regarding the training time and computational power. In fact, we made it a point to ensure that the additional diffusion pretraining step does not introduce significant overhead.
>
> We employed latent diffusion, which reduces the input to the diffusion model by a factor of 48, instead of having the diffusion process in the image space to reduce the memory and time required for training. The latent encoder that we used (TAESD) was specifically chosen given its compact size as well. This is despite the availability of other better latent encoders such as Stable Diffusion’s VAE.
>
> Moreover, as mentioned in our paper, currently there are no existing pretrained diffusion models for gait silhouette generation, and we had to resort to training the diffusion model from scratch together with the gait recognition model backbone. That is why we made an effort to ensure that the size of the diffusion model is not too large even though a larger denoiser is shown to provide a better improvement on the downstream task (as shown in Table 3). This is to minimize any additional overhead in training. Should there be a publicly available pretrained diffusion model for gait silhouette generation, we would not have to train the entire diffusion model, which will lower the computational demand and pretraining duration of our approach.  Our findings might be more useful and relevant in the near future when a pretrained diffusion model for gait silhouette generation is available for public use.
>
> As for deployment, do kindly note that our approach does not introduce any additional computational resource demand as only the finetuned gait recognition model will be used for gait recognition during deployment time.

---

> ### Author Response · Authors · 2024-11-19
> **Official Response to Reviewer pSRd (Part 2 of 2)**
>
> **Comment 3**: *The model's reliance on well-segmented and aligned input data for effective pretraining may reduce its robustness in truly uncontrolled environments where such quality cannot be guaranteed. This dependency could undermine the practical utility of the model in real-world applications where ideal data conditions are not met. A deeper investigation into the model's performance with lower quality or adversarially perturbed inputs would be valuable.*
>
> We understand your concern about the robustness of our approach. That is why in our study we decided to focus our efforts on gait recognition in the wild, where factors such as temporary occlusions, varying camera viewpoints, and illumination inadvertently come into play. In our study, we used two in-the-wild gait datasets – Gait3D and GREW.
>
> To further ensure robustness, we introduced data augmentation to both the inputs of the gait recognition model and diffusion model. This include RandomAffine, RandomPerspective, RandomHorizontalFlip, RandomPartDilate, and RandomPartBlur. As we are unable to include any images in the comments, we have added an additional appendix section that visualizes the effects of each augmentation technique applied (Appendix A.2). Note that the actual input seen by the model will be a composition of these augmentation techniques. These augmentation techniques have been found to enhance the generalizability of gait recognition models, which makes them more robust to the challenges posed in a truly unconstrained environment. Moreover, for the diffusion model, the forward diffusion process acts as an additional layer of augmentation where the inputs are further augmented with Gaussian noise.
>
> **Question 1**: *What are the computational costs associated with diffusion pretraining, and what strategies might be considered to reduce them?*
>
> Currently, diffusion pretraining involves training both the diffusion model and the gait recognition model backbone. As mentioned in the responses to **Comment 2**, in order to minimize this overhead, we have decided to employ latent diffusion which reduces the input to the diffusion model by a factor of 48. This helped to greatly reduce the memory and computational requirements. Furthermore, instead of simply going for a huge diffusion model, we opted for a medium-sized one that performed relatively well. Considering all these, we managed to train both our diffusion model and gait recognition model backbone with a single GPU. In fact, if you take a look at Table 7 in the appendix we provided, the majority of our diffusion pretraining experiments can fit within a single 24GB RTX 3090 GPU even with a batch size of 128 and sequence length of 30.
>
> Should there be a pretrained diffusion model for gait silhouette generation in the future, we would be able to avoid training the diffusion model, which will help to reduce the total training iterations required for diffusion pretraining and the computational resource required since we can choose to freeze the diffusion model parameters during diffusion pretraining.
>
> **Question 2**: *Could the authors elaborate on the potential applications of this method beyond gait recognition, perhaps in other areas of facial or emotion recognition?*
>
> As mentioned in our response to **Comment 1**, we believe our study can be extended to other fields that are largely similar to gait recognition such as speaker, facial, and emotion recognition. Many existing approaches in these fields adopt a similar model architecture and utilize the same loss function for training. Our current study provides a strong motivation for us and other researchers to venture into these fields to investigate if similar findings can be observed. Hopefully, this will also spur more research into leveraging diffusion models for representation learning.
>
> We hope that we have addressed all, if not most, of your current concerns. Nonetheless, if you still have any other doubts about our paper, feel free to reach out to us and we would be more than happy to clarify them. If all your concerns have been addressed, we would greatly appreciate it if you could reconsider the rating of our paper.

---

### Author Response · Authors · 2024-12-04
**Appreciation to all Reviewers**

We would like to express our sincere gratitude to all reviewers for their thoughtful and constructive feedback on our paper 'Diffusion Pretraining for Gait Recognition in the Wild'. We appreciate your time and effort in thoroughly evaluating our submission and suggesting improvements. We are glad that many of you have found our experiments comprehensive and the idea interesting. We hope the discussion phase has managed to clear up any major doubts you have about our work and led to a better understanding and appreciation of our work. We will keep all the suggestions that were brought up in mind and incorporate them into our future studies.

---

### Note · Authors · 2025-02-02

I have read and agree with the venue's withdrawal policy on behalf of myself and my co-authors.